# Equivariant Mesh Attention Networks

**Sourya Basu**[*]                                                                      *sourya@illinois.edu*
*University of Illinois at Urbana-Champaign, USA*

**Jose Gallego-Posada**[*]                                                              *gallegoj@mila.quebec*
*Mila and DIRO, Université de Montréal, Canada*

**Francesco Viganò**[*]                                                                 *f.vigano21@imperial.ac.uk*
*Imperial College London, UK*

**James Rowbottom**[*]                                                                  *jabrowbottom@gmail.com*
*Independent Scholar*

**Taco Cohen**                                                                          *tacos@qti.qualcomm.com*
*Qualcomm AI Research, The Netherlands*[†]

**Reviewed on OpenReview:** *https://openreview.net/forum?id=3IqqJh2Ycy*

## Abstract

Equivariance to symmetries has proven to be a powerful inductive bias in deep learning research. Recent works on mesh processing have concentrated on various kinds of natural symmetries, including translations, rotations, scaling, node permutations, and gauge transformations. To date, no existing architecture is equivariant to *all* of these transformations. In this paper, we present an attention-based architecture for mesh data that is provably equivariant to all transformations mentioned above. Our pipeline relies on the use of *relative tangential features*: a simple, effective, equivariance-friendly alternative to raw node positions as inputs. Experiments on the FAUST and TOSCA datasets confirm that our proposed architecture achieves improved performance on these benchmarks and is indeed equivariant, and therefore robust, to a wide variety of local/global transformations.

## 1 Introduction

Equivariance to symmetries has proven to be a powerful inductive bias in machine learning tasks ranging across classification, regression, segmentation, and reinforcement learning. Commonly studied symmetries include permutations (Keriven & Peyré, 2019; Zaheer et al., 2017), rotations (Cohen & Welling, 2016; Weiler et al., 2018; Li et al., 2018; Veeling et al., 2018), translations (LeCun et al., 1998), scaling (Sosnovik et al., 2020), and gauge transformations (Cohen et al., 2019a; de Haan et al., 2021).

These symmetries lead to different requirements for invariance or equivariance, depending on the learning task. For example, in a mesh shape-classification task, we would like to find a classifier that consistently predicts the same label, *invariant* to whether the input mesh is rotated, scaled or translated. In a node segmentation task, in addition to these invariances, we expect that the model predictions change in concurrence with a relabelling of the nodes, thus being permutation *equivariant*.

Achieving models with guaranteed equivariance is of high practical interest. The equivariance paradigm constitutes a principled way of incorporating task-specific prior information, which allows the model to simultaneously handle "equivalence classes of inputs" (i.e., those inputs related by symmetry transformations), and removes the need for data augmentation during training (Bronstein et al., 2021; Cohen, 2021).

---

[*]Equal contribution. Our code is available at: `https://github.com/gallego-posada/eman`
[†]Qualcomm AI Research is an initiative of Qualcomm Technologies, Inc.

In this work we concentrate on learning tasks involving 3-dimensional meshes as inputs. The main challenge in realizing equivariance in the case of mesh data lies in *adequately* handling the (arbitrary) numerical representation associated with the "geometric components" of a mesh. This is because different symmetry transformations have different effects on the mesh components. For example, a rotation modifies the positions of the nodes but leaves the face areas intact, while scaling leaves angles between nodes unchanged, but affects the positions, edge lengths and face areas.

While it is possible to discard the geometric structure of the mesh and consider it as a mere graph or point cloud, retaining this geometric information can be crucial for successful learning. Verma et al. (2018) and de Haan et al. (2021) report poor performance for models which discard connections between nodes (e.g. edges or faces). When used on mesh data, mesh-specific designs outperform models built for less-structured data, like 3D point clouds or embedded graphs. However, employing expressive models on meshes may require the introduction of additional constraints, such as equivariance with respect to gauge transformations (de Haan et al., 2021).

To achieve equivariance, the model predictions should depend *only* on the intrinsic geometry of the mesh, and not on the particular embedding used to represent the mesh computationally. In response to this challenge, the first contribution of our work is the introduction of **relative tangential** (RELTAN) **features**. RELTAN features transform absolute node positions into *tangent* vectors, in a way that accounts for the local geometry of the mesh at each node. We prove that this map is equivariant under global rotations, and invariant under translations and scaling of the ambient space $\mathbb{R}^3$.

Moreover, RELTAN features are *geometric features* (§3.2): although the numerical representation of these features may change depending on the specific choice of gauge, the geometric quantities represented by the features remain unchanged. We leverage the gauge equivariant convolutions of de Haan et al. (2021) as a building block to satisfy the gauge equivariance requirement.

The conjunction of RELTAN features and gauge equivariant convolutions allows us to design models that exhibit equivariance/invariance to **all** the symmetry transformations discussed above. Besides these desirable equivariance properties, our experiments demonstrate that relative tangential features provide consistent performance improvements. Thus, RELTAN features constitute a simple (yet effective!) alternative to using "raw" node positions as inputs.

Our second contribution is an extension of mesh processing algorithms to include a **gauge equivariant attention mechanism** in a way that provably satisfies the aforementioned requirements. We refer to this architecture as an Equivariant Mesh Attention Network (EMAN). Attention mechanisms have been a groundbreaking innovation in deep learning powering state-of-the-art results in natural language processing (Vaswani et al., 2017; Devlin et al., 2019; Brown et al., 2020), computer vision (Dosovitskiy et al., 2021), reinforcement learning (Chen et al., 2021), multi-modal learning (Jaegle et al., 2021), and graph neural networks (Veličković et al., 2018; Chamberlain et al., 2021). However, attention-based methods remain largely unexplored in the context of mesh data.

We carry out experiments on the FAUST (Bogo et al., 2014) and TOSCA (Bronstein et al., 2008) datasets. We do *not* apply any form of data augmentation during training. However, we evaluate the performance of all models on the test set, while applying transformations to the unseen examples. These transformations consist of rotations, translations, scalings, permutations and local gauge changes. We apply the transformations "one-at-a-time" as an ablation study which allows us to systematically verify the equivariance properties of all the compared models. Our results confirm that our proposed model design is the only one to achieve equivariance, and therefore robustness, to all of these local/global transformations

## 2   Related Work

**Equivariance on graphs and manifolds.** Equivariance on graphs and manifolds has been studied from numerous perspectives. Cohen et al. (2018; 2019b) propose novel gauge equivariant convolution techniques on manifolds such as spheres and platonic solids. Satorras et al. (2021) propose E($n$)-equivariant graph neural network models, which are applied to equivariant normalizing flows by Köhler et al. (2020). Moreover, de Haan et al. (2021) point out a relation between different types of equivariances: their gauge equivariant

model is also equivariant to the group of isometries of the given mesh. The equivariance with respect to this group of isometries has been extensively studied by Weiler et al. (2021) and Cohen (2021). In contrast, our work focuses on equivariance with respect to gauge transformations *and* global transformations of $\mathbb{R}^3$.

**Equivariant attention.** Attention has been used to provide more expressive filters over the isotropic convolutions in GCN (Kipf & Welling, 2017). Veličković et al. (2018); Shi et al. (2021) and Chamberlain et al. (2021) implement attention mechanisms to give anisotropic filters. Several works propose alternative forms of equivariant attention. Hutchinson et al. (2021) introduce an attention architecture that is equivariant to Lie group actions. SE(3)-Transformer (Fuchs et al., 2020) is a roto-translation equivariant attention network, designed for point-cloud data and graphs, and not meshes. Wang et al. (2020) propose a self-supervised equivariant attention mechanism in image segmentation that generates more consistent class activation maps over rescaling. Romero & Cordonnier (2021) propose a general group-equivariant self-attention formulation for processing images.

**Equivariant attention on meshes.** Gauge Equivariant Transformers (GET), recently proposed by He et al. (2021), also consider gauge equivariant attention on meshes. The authors propose features which represent the raw node positions in the local coordinate system of each node. We refer to these as GET features. These features are equivariant to global rotations, but are not equivariant to scaling or translation of the mesh. In contrast, RELTAN features are local features which successfully provide invariance to scaling and translations, in addition to equivariance to global rotations (Section 4).

Moreover, the attention mechanism used in GET is only gauge equivariant to multiples of $2\pi/N$, for a certain positive integer hyperparameter $N$. Only when $N$ is odd, He et al. (2021) provide a framework for *approximate* equivariance to arbitrary gauge transformation, along with an estimate of the approximation error incurred. In contrast, EMAN attention mechanism is *exactly* equivariant to arbitrary transformations of gauge (Section 6). A more detailed discussion about the difference between our architecture and choice of input features and those used in GET is provided in Appendix H.

## 3 Geometry

In this section we describe meshes, geometric features, parallel transport, equivariances and invariances and graph convolutions. Familiar readers may choose to proceed directly to Section 4.

### 3.1 Meshes

A ***mesh*** $\mathcal{M}$ is determined by a set of vertices, or nodes, a set of (undirected) edges, and a set of faces. We consider oriented meshes embedded in the ambient space $\mathbb{R}^3$ and require the faces to "properly glue along the edges", so that $\mathcal{M}$ is in fact a piece-wise linear sub-manifold of $\mathbb{R}^3$. One can think of the mesh $\mathcal{M}$ as a discretization (e.g., a triangulation) of a 2-dimensional manifold.

A 2-dimensional sub-manifold of $\mathbb{R}^3$ admits a ***tangent plane*** at each point. The discrete equivalent for a mesh $\mathcal{M}$, at a point $p \in \mathcal{M}$, is the plane orthogonal to the normal vector

$$n_p = \frac{\sum_{F \ni p} \mathcal{A}(F) \, n_F}{|| \sum_{F \ni p} \mathcal{A}(F) \, n_F ||}, \tag{1}$$

where $|| \cdot ||$ denotes the 2-norm. The normal vector $n_p$ (at a point $p$) is computed as a weighted sum of the normal vectors to the adjacent faces $\{F \ni p\}$, where the contribution of face $F$ is proportional to its area $\mathcal{A}(F)$. We denote the tangent plane at a point $p$ by $T_p\mathcal{M}$.

For the tangent plane $T_p\mathcal{M}$ at $p$, we consider a ***gauge***, or frame, $\mathbf{E}_p = \{e_{p,1}, e_{p,2}\}$. We only allow orthonormal gauges, for which the triple $\mathbf{E}_p \cup \{n_p\}$ constitutes a positively oriented orthonormal basis of $\mathbb{R}^3$. Therefore, any other *admissible* orthonormal frame of $T_p\mathcal{M}$ is obtained from the previous gauge $\mathbf{E}_p$ by a rotation $g \in \mathrm{SO}(2)$. We use $g$ to denote both the rotation and its corresponding angle, modulo $2\pi$.

Finally, we define angles $\theta_{pq}$, which take into account the local orientation of the neighbors of a node $p$. $\theta_{pq}$ is the angle between the projection of the vector $q - p$ onto $T_p\mathcal{M}$ and the reference axis $e_{p,1}$. The angles $\theta_{pq}$ are thus gauge-dependent quantities.

### 3.2 Geometric Features

***Geometric features*** are a central concept in our work. Here we closely follow the presentation and notation of de Haan et al. (2021). Meshes possess a richer structure than mere graphs. An important insight in geometric deep learning is to take the mesh structure into consideration, and allow the features on the underlying space to be not just simple *functions* on the space, but rather *sections of vector bundles.*

As an example, a tangential feature $f$ on a mesh is given by a choice of tangent vector in the plane $T_p\mathcal{M}$, for each point $p \in \mathcal{M}$. The tangent vector $f(p)$, determined by evaluating $f$ at $p$, can be represented by a 2-dimensional vector of coordinates $f_p$ with respect to the gauge $\mathbf{E}_p$. Note that the coordinate vector $f_p \in \mathbb{R}^2$ is *dependent* on the choice of a gauge, while the tangent vector $f(p) \in T_p\mathcal{M}$ is independent.

Therefore, as we change the gauge at $p$, we should *prescribe* how the coordinate vector $f_p$ gets modified. For tangential features, $f_p$ changes as $f_p \mapsto \rho_1(-g)f_p$, where $\rho_1(-g)$ denotes the rotation by the angle $-g$. This transformation rule precisely characterizes tangential features. In contrast, scalar features are 1-dimensional features that *do not change* when the gauge is transformed, and thus we may write $f_p \mapsto \rho_0(-g)f_p$, where $\rho_0(-g) = 1$ for all $g \in \mathrm{SO}(2)$.

More generally, for $n \in \mathbb{N}$, $n \geq 1$, we can consider 2-dimensional features $f$, that change $f_p \mapsto \rho_n(-g)f_p$ under a gauge change $g \in \mathrm{SO}(2)$. Here $\rho_n(-g)$ denotes the rotation by angle $-ng$. The ***representations*** $\rho_n$ are

$$\rho_0(g) = 1, \quad \rho_n(g) = \begin{pmatrix} \cos ng & -\sin ng \\ \sin ng & \cos ng \end{pmatrix} \text{ for } n \geq 1.$$

We say that a feature is of ***type*** $\rho$ if it changes accordingly with the representation $\rho$. The $\{\rho_n\}$ form an exhaustive list of irreducible representations, the building blocks of all finite-dimensional representations of $\mathrm{SO}(2)$. In other words, there is no loss of generality in considering *only* geometric features corresponding to direct sums of such representations, obtained by concatenation of multiple irreducible components. We consider these types of features throughout the rest of the paper. For example, $\rho = 4\rho_0 \oplus \rho_1 \oplus 3\rho_2$ corresponds to a feature type of dimension $4 \cdot 1 + 1 \cdot 2 + 3 \cdot 2$. Note also that these feature types are *orthogonal*, meaning that $\rho^\top = \rho^{-1}$.

### 3.3 Parallel Transport

Message passing updates involve processing features stored at different nodes of the mesh. However, geometric features present a challenge. For instance, tangential features stored at different nodes belong to different tangent planes (i.e. different vector spaces), and thus are not immediately comparable. ***Parallel transport*** is a procedure from differential geometry that describes how to "coherently translate" between tangent planes at different points, respecting the curvature of the manifold.

Discrete parallel transport can be intuitively understood as follows[*]: for a node $p \in \mathcal{M}$, and a neighbor $q \in \mathcal{N}_p$, we first translate the tangent plane $T_q\mathcal{M}$, together with the normal vector $n_q$, to $p$, along the edge joining $q$ and $p$. Then, we consider the unique rotation of $\mathbb{R}^3$ that maps $n_q$ to $n_p$, with fixed axis $n_q \times n_p$. Under this rotation, $T_q\mathcal{M}$ is mapped onto $T_p\mathcal{M}$, and it is now coherent to *compare tangent vectors* at $q$ with tangent vectors at $p$.

However, a feature $f$ on $\mathcal{M}$ is represented by coordinate vectors $f_p$ and $f_q$ *with respect to two different gauges.* In general, even after rotating $T_q\mathcal{M}$ onto $T_p\mathcal{M}$, the two gauges do not coincide. Therefore, we denote by $-g_{q \to p} \in \mathrm{SO}(2)$ the 2-dimensional rotation corresponding to the gauge change from the given gauge at $q$ (after rotating $T_q\mathcal{M}$ onto $T_p\mathcal{M}$), and the given gauge at $p$. It is now coherent to *compare coordinate vectors* $f_p$ with $\rho(g_{q \to p})f_q$, as they both represent coordinates of geometric features of the same type $\rho$, at the same point $p$, *with respect to the same gauge* $\mathbf{E}_p$. If we denote by $R_{q \to p}$ the unique rotation of $\mathbb{R}^3$ that maps $n_q$ to $n_p$, with fixed axis $n_q \times n_p$, we can express (the angle) $g_{q \to p}$ as:

$$g_{q \to p} = \mathrm{atan2}\big((R_{q \to p}e_{q,2})^\top e_{p,1}, (R_{q \to p}e_{q,1})^\top e_{p,1}\big). \tag{2}$$

---

[*]See Fig. 3 in de Haan et al. (2021) for a nice illustration.

### 3.4 Equivariances and Invariances

Throughout this section, we denote by $\mathcal{F}_{\text{in}}$ and $\mathcal{F}_{\text{out}}$ the spaces of features of type $\rho_{\text{in}}$ and $\rho_{\text{out}}$, respectively, and by $\mathcal{F}$ a generic space of features of type $\rho$. Also, given a transformation $\Psi$ applied on a mesh $\mathcal{M}$, we use $\Psi_*$ to denote the pushforward operator induced by $\Psi$. We describe specific cases of this pushforward below.

**Gauge equi/invariance.** To coherently define a feature mapping $\mathcal{K}\colon \mathcal{F}_{\text{in}} \to \mathcal{F}_{\text{out}}$, we require its computation to be independent of the choice of the gauge. Consider an arbitrary change of gauge $g \in \mathrm{SO}(2)$. Since features of type $\rho_{\text{in}}$ transform as $f_p \mapsto \rho_{\text{in}}(-g)f_p$, and similarly for $\rho_{\text{out}}$, we demand $\mathcal{K}$ to satisfy the **gauge equivariance** constraint

$$\rho_{\text{out}}(-g) \circ \mathcal{K} = \mathcal{K} \circ \rho_{\text{in}}(-g). \tag{3}$$

If the representation $\rho_{\text{out}}$ is a (direct sum of) scalar feature(s) $\rho_0$, then we talk about **gauge invariance**, as the resulting features are insensitive to the particular choice of gauge.

**Global rotation equi/invariance.** Given a gauge-equivariant feature mapping $\mathcal{K}\colon \mathcal{F}_{\text{in}} \to \mathcal{F}_{\text{out}}$, we study its interaction with a global rotation $R \in \mathrm{SO}(3)$. Denote by $R\mathcal{M}$ the mesh obtained by rotating $\mathcal{M}$. We write $\mathcal{K}^{\mathcal{M}}\colon \mathcal{F}_{\text{in}}^{\mathcal{M}} \to \mathcal{F}_{\text{out}}^{\mathcal{M}}$ and $\mathcal{K}^{R\mathcal{M}}\colon \mathcal{F}_{\text{in}}^{R\mathcal{M}} \to \mathcal{F}_{\text{out}}^{R\mathcal{M}}$, to distinguish between *the same feature mapping* applied to feature spaces *on two different meshes.*

If $f \in \mathcal{F}^{\mathcal{M}}$, the rotation $R$ transforms $f$ to a feature $R_* f \in \mathcal{F}^{R\mathcal{M}}$. Formally, if $f$ is represented at $p$ by the coordinate vector $f_p$ relative to the gauge $\mathbf{E}_p$, then $R_* f$ is represented at $Rp$ by the same coordinate vector $(R_* f)_{Rp} = f_p$, relative to the *rotated* gauge $R\mathbf{E}_p$. For example, when $f_p$ is a tangent vector $f_p \in T_p\mathcal{M}$, the action of $R_*$ corresponds geometrically to the rotation of the vector $f_p$ by $R$.

*Global rotation equivariance* of the feature mapping $\mathcal{K}$ is given by:

$$R_* \circ \mathcal{K}^{\mathcal{M}} = \mathcal{K}^{R\mathcal{M}} \circ R_*. \tag{4}$$

Again, if the representation $\rho_{\text{out}}$ is one, or a sum of scalar features $\rho_0$, then we talk about **global rotation invariance**.

**Global translation invariance.** A translation $Tp = p + x$ of $\mathbb{R}^3$, with $x \in \mathbb{R}^3$, trivially pushes features on $\mathcal{M}$ to features on $T\mathcal{M}$: if $f \in \mathcal{F}^{\mathcal{M}}$ is represented by $f_p$ in the gauge $\mathbf{E}_p$ at $p$, then $T_* f \in \mathcal{F}^{T\mathcal{M}}$ is represented by the same coordinate vector $(T_* f)_{Tp} = f_p$ in the same gauge $\mathbf{E}_p$ at $Tp$. Intuitively, $T_* f$ is the same feature as $f$, "stored" at the translated points. Therefore, we say that a feature mapping $\mathcal{K}$ is **global translation invariant** if

$$T_* \circ \mathcal{K}^{\mathcal{M}} = \mathcal{K}^{T\mathcal{M}} \circ T_*. \tag{5}$$

**Global scaling invariance.** A scaling $Sp = \lambda p$ of $\mathbb{R}^3$, $\lambda > 0$, similarly allows a definition of a push-forward $S_*\colon \mathcal{F}^{\mathcal{M}} \to \mathcal{F}^{S\mathcal{M}}$: if $f \in \mathcal{F}^{\mathcal{M}}$ is represented by $f_p$ in the gauge $\mathbf{E}_p$ at $p$, then $S_* f_p \in \mathcal{F}^{S\mathcal{M}}$ is represented by the same coordinates $(S_* f)_{Sp} = f_p$ in the gauge $\mathbf{E}_p$ at $Sp$.

While for rotations $R$ and translations $T$ new gauges are obtained through the differentials $dR$ and $dT$, this is not the case for scalings $S$, since the new gauge would not be normalized. With our definition, tangential features *do not scale* as the mesh scales, and preserve their norm. For this reason, we say that a feature mapping $\mathcal{K}$ is **global scaling invariant** (and not equivariant) if

$$S_* \circ \mathcal{K}^{\mathcal{M}} = \mathcal{K}^{S\mathcal{M}} \circ S_*. \tag{6}$$

**Node permutation equi/invariance.** Consider a re-labeling of the nodes in a mesh $\mathcal{M}$. We talk about **node permutation equivariance** (resp. **node permutation invariance**) when the outcome of a process computed over rearranged nodes is the rearrangement of the outcome from the original ordering, under the same permutation (resp. does not depend on the ordering of the nodes). We expect a segmentation model to be permutation equivariant, while a classification model should be permutation invariant.

### 3.5 Graph Convolution on Meshes

If we ignore the faces of a mesh and its embedding in $\mathbb{R}^3$, we obtain a graph. The works of Scarselli et al. (2009); Bruna et al. (2013); Defferrard et al. (2016) and Kipf & Welling (2017) led to Graph Convolutional

Networks (GCNs), which give an efficient algorithm for node classification. However, GCNs do not account for the local geometry of the mesh and use the same *isotropic* kernel to process signals from all neighbors.

Gauge Equivariant Mesh CNNs (GEM-CNNs) (de Haan et al., 2021) were introduced to overcome this geometric obstacle. Their update step uses **anisotropic** kernels that depend on the spatial arrangement of the neighboring nodes $\{q \in \mathcal{N}_p\}$. The message passing in a GEM-CNN is performed as

$$f'_p = K_{\text{self}} f_p + \sum_{q \in \mathcal{N}_p} K_{\text{neigh}}(\theta_{pq}) \rho_{\text{in}}(g_{q \to p}) f_q. \tag{7}$$

The kernel $K_{\text{neigh}}(\theta)$ depends on the angle $\theta_{pq}$ formed by the edge $p \to q$, with respect to the reference gauge at $p$. Moreover, the kernels $K_{\text{self}}$ and $K_{\text{neigh}}(\theta)$ satisfy geometric constraints, so that the output feature $f'_p$ transforms accordingly with the change of gauge. Hence, GEM-CNNs take into account the local geometry of the mesh, while also ensuring *equivariance* to change of gauge. In particular, $K_{\text{self}}$ and $K_{\text{neigh}}(\theta)$ satisfy:

$$K_{\text{self}} = \rho_{\text{out}}(-g) \ K_{\text{self}} \ \rho_{\text{in}}(g), \qquad K_{\text{neigh}}(\theta - g) = \rho_{\text{out}}(-g) \ K_{\text{neigh}}(\theta) \ \rho_{\text{in}}(g). \tag{8}$$

For details on $K_{\text{self}}$ and $K_{\text{neigh}}(\theta)$, please see Appendix A, and de Haan et al. (2021).

## 4 Relative Tangential Features

Given a mesh $\mathcal{M}$, we construct **relative tangential** (RELTAN) **features** $v_p(r)$, depending on the local geometry of the mesh $\mathcal{M}$ around a node $p$. We use the adjective *relative* to underline their dependency on the *relative* node positions $q - p$ of the neighbors $\{q \in \mathcal{N}_p\}$, rather than the *absolute* node positions. As shown in Lemma 4.1, RELTAN features provide global rotational equivariance, and invariance under translations and scaling of the ambient space $\mathbb{R}^3$. At a node $p$, we define the 3-dimensional vector $v_p$ as:

$$v_p(r) = \frac{1}{N_p^{3/2}} \sum_{q \in \mathcal{N}_p} \pi_p \left( \frac{q-p}{||q-p||} \right) \cdot \left[ \frac{||q-p||^{r-1}}{\sum_{q' \in \mathcal{N}_p} ||q'-p||^{r-1}} \right]^{-1}$$

where $N_p = |\mathcal{N}_p|$ denotes the degree of node $p$, and the projection $\pi_p$ onto the tangent plane $T_p\mathcal{M}$ is $I - n_p n_p^\top$. The factor $1/N_p^{3/2}$ is included in order to guarantee a correct asymptotic behavior of $v_p(r)$ as the node degree $N_p$ increases; for a detailed discussion of this normalizing factor, see Appendix B.1.

RELTAN features provide a convenient geometric input, as they satisfy the following properties:

**Lemma 4.1.** *For any $r \in \mathbb{R}$, the process of computing relative tangential features $v_p(r)$ is equivariant under global rotations, and invariant under translations and scaling of $\mathbb{R}^3$. Namely, if $R \in SO(3)$ is a rotation, $x \in \mathbb{R}^3$ is a translation vector, and $\lambda > 0$ is a scaling factor, then* [Proof]

$$v_{Rp}^{R\mathcal{M}} = R(v_p^{\mathcal{M}}), \quad v_{\lambda p}^{\lambda \mathcal{M}} = v_p^{\mathcal{M}}, \quad v_{p+x}^{\mathcal{M}+x} = v_p^{\mathcal{M}}.$$

See Appendix B.2 for a proof of this result. Thanks to their geometric properties, RELTAN features constitute a simple, equivariance-friendly alternative to using raw node positions as input features.

**Influence of the relative power $r$.** Each of the neighbors $q \in \mathcal{N}_p$ affects the computation of $v_p(r)$ in two ways. First, there is a directional component $\pi_q \left( \frac{q-p}{||q-p||} \right)$ which considers the alignment between the edge connecting $q$ to $p$ and the tangent plane at $p$. Second, the distances between all neighbors and $p$ are used to weigh the directional contributions: neighbors with *smaller* distances contribute *more*.

Note that the effect of the distances is mediated by the power $r - 1$. We refer to the real-valued parameter $r$ as the **relative power**. As the relative power $r$ decreases, the contributions of far-away neighbors are highlighted. In contrast, as the relative power increases, neighbors close to $p$ become most relevant in the computation of $v_p(r)$. In particular, when $r = 1$, the distances are ignored and only the directional components affect the final value of $v_p(r)$. These behaviors are illustrated in Fig. 1.

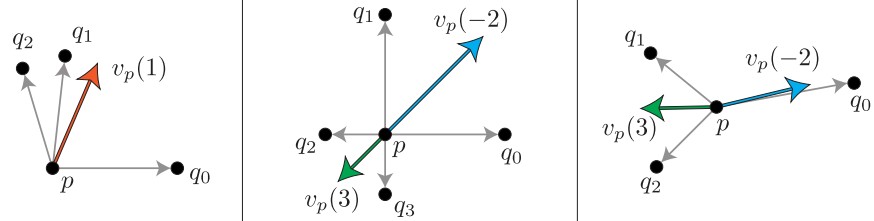

Figure 1: Examples of computation of RELTAN features on planar neighborhoods. Orange, blue and green vectors represent relative tangent features $v_p(r)$ for $r = 1$, $r = -2$ and $r = 3$, respectively. For small relative powers $r$, neighbors far from $p$ contribute more to the relative tangent feature $v_p(r)$, and vice versa.

**Selecting relative powers.** As discussed above, different values of $r$ provide different perspectives on the local geometry of the mesh $\mathcal{M}$ around the node $p$. Balancing the importance of the directional and distance components may depend on domain-specific properties of the data. Moreover, multiple relative powers can be simultaneously used for capturing information of the local neighborhoods "at different scales". Which of these scales is most relevant for the task at hand can be in turn learned as part of the optimization of the model weights during training. Note, however, that this strategy increases the number of parameters in the model. Hence, one should use enough relative powers that can capture rich information about the nodes while not being computationally wasteful. In our experiments, choosing two relative powers simultaneously provided desirable performance.

## 5 Verifiably Equivariant Message Passing

In this section we empirically verify that RELTAN features, coupled with a suitable choice of bias for the convolutional layers, allow us to build models that are in fact equi/invariant to all the transformations mentioned in Section 3.4. We highlight the effect that "small" design choices, such as biases, can have when trying to integrate them towards building a fully equivariant pipeline. We also emphasize on the importance of performing a thorough evaluation of the model by applying the transformations of interest to unseen inputs in order to reliably verify the desired equivariance properties.

**Designing equivariant biases.** The traditional way of including biases in standard convolutional layers involves the addition of a fixed vector across the different channels of the output tensor. However, in the context of mesh data, this procedure is *not* equivariant to changes of gauge. When considering geometric features, the addition of this "fixed" bias vector would correspond to summing a gauge-sensitive quantity to the coordinate vectors representing a feature. As a response to this, we consider *angular biases* that respect gauge equivariance.

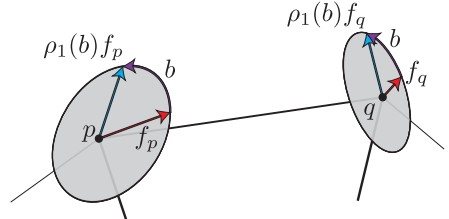

Figure 2: Angular bias applied to the tangential features in the convolution blocks. Original features are coloured in red, bias in purple, new features in blue. Equivariance is preserved by *rotating* $\rho_{n \geq 1}$-features.

Given a general representation $\rho$, we decompose it in its irreducible components $\{\rho_n\}$. The cumulative bias for $\rho$ is assembled from the biases on its irreducible components. If $n = 0$, we may add a simple (non-angular) bias $b$ to the 1-dimensional scalar feature $f$. For $n > 0$, we instead rotate the 2-dimensional coordinate vector $f_p$ by an angular bias $b$, or equivalently we consider $\rho_n(b)f_p$. See Section 5. Therefore, for a feature of type $\rho$, the number of involved biases is the same as the number of irreducible components in $\rho$. This choice of bias is therefore suitable for our goal of designing a fully equivariant model.

**Verifying model equivariance.** Table 1 compares the violation of equivariance to global/local transformations exhibited by randomly initialized models.[†] We consider a SpiralNet++ (Gong et al., 2019) using raw

---

[†]We use untrained models since we are interested in assessing the equivariance of the model, rather than its accuracy.

positions as inputs; and a GEM-CNN model (de Haan et al., 2021) using raw positions, RELTAN features and GET features (He et al., 2021) as inputs, as well as the non-equivariant, and angular biases mentioned above. GET features correspond to raw node positions, represented using the local coordinate system at each node. A detailed description of GET features is given in Appendix H. Since we are interested in obtaining a model that is equivariant to arbitrary gauge changes, and not only to multiples of a given rotation, we do not employ (quantized) regular non-linearities (de Haan et al., 2021, §4).

A GEM-CNN model with RELTAN features and angular bias is the *only* configuration that achieves equivariance to *all* the considered transformations. Note that SpiralNet++ is perfectly gauge equivariant as it only employs scalar features. The larger magnitude observed for the minimum error achieved for gauge transformations is explained by the fact that gauge transformations affect *every* convolutional layer of the model, causing numerical errors to accumulate.

Table 1: Equivariance gap of randomly initialized models for various transformations on FAUST meshes. The gap is computed as the MSE between the logits for the outputs corresponding to the *same* mesh with or without a given transformation. The reported value is an average across all test meshes. Entries with small (resp. high) error are shown in blue (resp. red). Values in this table should be compared within the same column, based on the order of magnitude of the realized errors.

| Model | Bias | Initial Features | Equivariance Gap | | |
| | | | Gauge | Rot-Tr-Scale | Perm |
|---|---|---|---|---|---|
| **SpiralNet++** | - | XYZ | 0 | $1.03 \cdot 10^{-1}$ | 0 |
| **GEM-CNN** | Non-Equiv | XYZ | $1.43 \cdot 10^{-1}$ | $6.55 \cdot 10^{-1}$ | $2.26 \cdot 10^{-13}$ |
| | Angular | XYZ | $7.95 \cdot 10^{-6}$ | 1.19 | $1.55 \cdot 10^{-13}$ |
| | | GET | $8.75 \cdot 10^{-6}$ | 2.60 | $1.69 \cdot 10^{-13}$ |
| | | RELTAN | $1.31 \cdot 10^{-5}$ | $5.57 \cdot 10^{-9}$ | $1.88 \cdot 10^{-13}$ |

**The importance of tansformations in *evaluation*.** We complete this section by studying the robustness of a similar set of models *after training* on the FAUST dataset. Experimental details can be found in Appendix E. We do not apply any transformations to meshes in the training set. The results displayed in Table 2 show that the pattern of robustness to these transformations carries out verbatim, from that of untrained models shown in Table 1.

Table 2: Accuracy for models trained on FAUST. No data augmentation is applied during training. The last 4 columns represent the performance of the model on the test set under *different transformations*. Blue entries show robustness to transformations for each column, whereas the red entries correspond to poor performance.

| Model | Bias | Initial Features | Accuracy (%) | | | | |
| | | | Train | Test | Gauge | Rot-Tr-Scale | Perm |
|---|---|---|---|---|---|---|---|
| **SpiralNet++** | - | XYZ | 100.0 | 99.91 | 99.91 | 0.30 | 99.91 |
| **GEM-CNN** | Non-Equiv | XYZ | 99.99 | 99.90 | 0.06 | 12.48 | 99.90 |
| | Angular | XYZ | 99.45 | 97.97 | 96.85 | 0.17 | 97.97 |
| | | GET | 99.18 | 97.75 | 97.40 | 0.61 | 97.75 |
| | | RELTAN[0.7] | 99.68 | 98.69 | 98.20 | 98.69 | 98.69 |
| | | RELTAN[0.5, 0.7] | 99.63 | 98.36 | 97.84 | 98.36 | 98.36 |

However, we highlight that the shortcomings of the non-equivariant models **cannot be detected** by looking *only* at the un-transformed training and test set accuracies! For example, an inadequate choice of a "small" component like the bias used in the convolutional layers can drastically affect the equivariance of the model: compare the equivariance to gauge transformations for a GEM-CNN model with (equivariant) angular bias and with (non-equivariant) additive bias. Therefore, when evaluating equivariance-aimed models, applying

the transformations of interest is crucial for successfully validating whether a model indeed satisfies the desired equivariance properties.

We do not include accuracies for the SpiralNet++ model with RELTAN features. However, one finds that this model with RELTAN features is invariant to translations and scalings, as RELTAN features are invariant to these transformations. Nonetheless, the SpiralNet++ model with RELTAN features is not global rotation equivariant, since its layers are not designed to be compatible with rotations of the ambient space $\mathbb{R}^3$. We included a short note on node permutation equivariance of the SpiralNet++ model in Appendix D.

## 6 Equivariant Mesh Attention Networks

Equipped with a "fully equivariant" pipeline comprising a base GEM-CNN model with angular biases and the use of RELTAN features, we now proceed to the presentation of our proposed attention mechanism.

The typical update step employed in GCNs considers the information coming from all the neighbors $q \in \mathcal{N}_p$ to be equally important when computing the update at node $p$. The similarity or alignment between the features $f_p$ and $f_q$ is irrelevant. Veličković et al. (2018) introduced Graph Attention Networks (GATs) to address this expressivity issue. In the update step, the message passed from neighbors is scaled using attention weights dependent on $f_p$ and $f_q$.

Equivariant Mesh Attention Networks (EMAN) are the second contribution of our work. EMAN combines ① anisotropic kernels (de Haan et al., 2021), with ② attention coefficients $\alpha_{pq}$ relating neighboring nodes (Veličković et al., 2018). The kernel is gauge equivariant, and the attention coefficients are scalar features (namely, unaffected by gauge transformations). This combination results in a gauge equivariant attention model. The convolutional update for EMAN is given by:

$$f'_p = \sum_{q \in \mathcal{N}_p} \alpha_{pq} \, K(\theta_{pq}) \rho_{\text{in}}(g_{q \to p}) f_q. \tag{9}$$

**Equivariant Mesh Attention Layers.** Algorithm 1 provides an overview of the design of our attention mechanism. The definitions of the quantities involved are presented below.

---

**Algorithm 1** Convolutional update in an Equivariant Mesh Attention Layer

**Forward** $((f_p)_{p \in \mathcal{M}}, K_{\text{query}}, K_{\text{key}}(\theta), K_{\text{value}}(\theta))$**:**
    **for** $p \in \mathcal{M}$**:**
        $\mathbf{Q}_p \leftarrow K_{\text{query}} f_p$
        $\mathbf{K}_p \leftarrow \texttt{Concatenate}(K_{\text{key}}(\theta_{pq}) \rho_{\text{in}}(g_{q \to p}) f_q \text{ for } q \in \mathcal{N}_p)$
        $\mathbf{V}_p \leftarrow \texttt{Concatenate}(K_{\text{value}}(\theta_{pq}) \rho_{\text{in}}(g_{q \to p}) f_q \text{ for } q \in \mathcal{N}_p)$
        $f'_p \leftarrow N_p \cdot \mathbf{V}_p \cdot \texttt{softmax}\left(\frac{\mathbf{K}_p^\top \mathbf{Q}_p}{\sqrt{C_{\text{att}}}}\right)$
**Output:** $(f'_p)_{p \in \mathcal{M}}$

---

We start by considering an auxiliary representation $\rho_{\text{att}} \colon \text{SO}(2) \to \mathbb{R}^{C_{\text{att}}}$. Let $K_{\text{query}}$ be a $C_{\text{att}} \times C_{\text{in}}$ matrix. Consider families $K_{\text{key}}(\theta)$ and $K_{\text{value}}(\theta)$ of matrices of size $C_{\text{att}} \times C_{\text{in}}$ and $C_{\text{out}} \times C_{\text{in}}$, respectively. Given a node $p$ on the mesh, for every neighbor $q \in \mathcal{N}_p$, we compute:

$$\mathbf{Q}_p = K_{\text{query}} f_p, \qquad \mathbf{K}_{pq} = K_{\text{key}}(\theta_{pq}) \rho_{\text{in}}(g_{q \to p}) f_q, \qquad \mathbf{V}_{pq} = K_{\text{value}}(\theta_{pq}) \rho_{\text{in}}(g_{q \to p}) f_q, \tag{10}$$

To provide gauge equivariance, we impose constraints on the matrices $K_{\text{query}}$, $K_{\text{key}}$ and $K_{\text{value}}$. These constraints must be respected for all $g \in \text{SO}(2)$. The solutions to these equations are provided in Appendix A.

$$\begin{aligned} K_{\text{query}} &= \rho_{\text{att}}(-g) \, K_{\text{query}} \, \rho_{\text{in}}(g), \\ K_{\text{key}}(\theta - g) &= \rho_{\text{att}}(-g) \, K_{\text{key}}(\theta) \, \rho_{\text{in}}(g) \\ K_{\text{value}}(\theta - g) &= \rho_{\text{out}}(-g) \, K_{\text{value}}(\theta) \, \rho_{\text{in}}(g), \end{aligned} \tag{11}$$

We define $\mathbf{K}_p$ and $\mathbf{V}_p$ as the $C_{\text{att}} \times N_p$ and $C_{\text{out}} \times N_p$ matrices obtained by concatenating as columns the vectors $\mathbf{K}_{pq}$, and $\mathbf{V}_{pq}$, respectively, over the neighbors $q \in \mathcal{N}_p$:

$$\mathbf{K}_p = \text{Concat}(\mathbf{K}_{pq} \text{ for } q \in \mathcal{N}_p), \qquad \mathbf{V}_p = \text{Concat}(\mathbf{V}_{pq} \text{ for } q \in \mathcal{N}_p). \tag{12}$$

The value of the updated feature $f'_p$ (of dimension $C_{\text{out}}$) is then given by:

$$f'_p = \sum_{q \in \mathcal{N}_p} \overbrace{\left[ N_p \cdot \text{softmax}\left( \frac{\mathbf{K}_p^\top \mathbf{Q}_p}{\sqrt{C_{\text{att}}}} \right) \right]_q}^{\alpha_{pq}} \underbrace{K_{\text{value}}(\theta_{pq})\rho_{\text{in}}(g_{q\to p})f_q}_{\text{as in GEM-CNNs}} = N_p \cdot \mathbf{V}_p \cdot \text{softmax}\left( \frac{\mathbf{K}_p^\top \mathbf{Q}_p}{\sqrt{C_{\text{att}}}} \right). \tag{13}$$

Note that the factor $N_p$ is not present in the work of Veličković et al. (2018). We introduce it here so that Eq. (13) precisely generalizes GEM-CNN convolution (with no self-contribution), in the case where the components of the softmax vector are all equal to $1/N_p$ (compare with discussion in Section 3.5).

**Equivariance properties of EMAN.** Thanks to the constraints satisfied by the kernels $K_{\text{self}}, K_{\text{key}}(\theta)$, and $K_{\text{value}}(\theta)$, we obtain gauge equivariance for EMAN (Lemma 6.1). This is illustrated in Fig. 3.

**Lemma 6.1.** *The convolutional update in Eq. (13) is gauge equivariant.* [Proof]

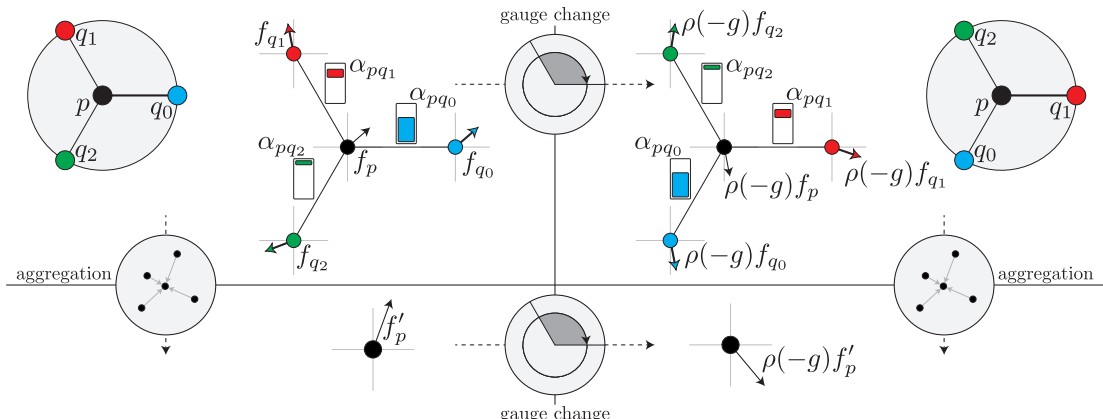

Figure 3: Message passing mechanism in Equivariant Mesh Attention Networks. For convenience, we represent a planar portion of the mesh and therefore ignore parallel transport. Tangent vectors $f_p, f_{q_i}$ are aggregated according to the attention coefficients $\alpha_{pq_i}$ (on the figure, going from top to bottom). A change of gauge reference neighbor (from $q_0$ to $q_1$) determines a rotation $\rho(-g)$ on tangent vectors of angle $-g$ (on the figure, going from left to right). Attention coefficients are invariant under gauge change. Pictorially, gauge equivariance can be rephrased as: "go right, go down" is the same as "go down, go right".

**Lemma 6.2.** *With the choice of kernels given by Eq. (11), the Equivariant Mesh Attention convolutional update in Eq. (13) (and thus Algorithm 1) is equivariant to global rotations, and invariant to translations and scalings of the ambient space $\mathbb{R}^3$.* [Proof]

The complete pipeline we propose in this work is formed by ① the use of RELTAN features (Section 4) as inputs, ② angular biases (Section 5) in the convolutional operators, and ③ EMAN layers (with the update rule from Algorithm 1). In conjunction, all these components contribute to a model architecture which is equivariant with respect to the complete range of transformations we are interested in:

**Theorem 6.3.** *With initial relative tangential features, Equivariant Mesh Attention Networks are equivariant to node permutations, and invariant under global rotations, translations and scalings of the ambient space $\mathbb{R}^3$, and under arbitrary gauge changes.* [Proof]

**Self-contribution.** The self-contribution $\alpha_{pp}K_{\text{self}}f_p$ is not present in Eq. (9). The base implementation of GEM-CNNs (de Haan et al., 2021) we used as a baseline did not include the self-contribution in the

convolutional step. To perform a fair comparison between our attention mechanism and GEM-CNNs, we do not include the self-contribution in our implementation either and leave experiments with self-contribution to future work. However, our formulation can be easily extended to include self-contributions. We present the details of this extension in Appendix C.5.

**Multi-head attention.** Our model can also support multi-head attention. See Appendix C.4 for details. We did not notice an improvement in performance when integrating this factor in our implementation for the considered tasks. The experimental results presented below consider single-head attention only.

## 7  Experiments

We carry out experiments on the FAUST (Bogo et al., 2014) and TOSCA (Bronstein et al., 2008) datasets for segmentation and classification tasks, respectively. We compare GEM-CNN and EMAN models using raw node XYZ positions, GET and RELTAN features as inputs. All the models use the angular biases described in Section 5. Details on our experimental settings can be found in Appendix E.

- No transformations are applied to the training meshes. All our experiments involving test accuracy report test results applying different transformations to the unseen meshes.

- Since we are interested in obtaining a model that is equivariant to arbitrary gauge changes, and not only to multiples of a given rotation, we do not employ (quantized) regular non-linearities (de Haan et al., 2021, §4).

- In Appendix F we provide performance comparison between equivariant models, and non-equivariant models trained using data augmentation. Appendix G contains a time-complexity comparison between GEM-CNN and EMAN.

**Comparison with GET** (He et al., 2021)**.** Both GEM-CNN and EMAN are equivariant to arbitrary gauge transformations unlike the GET model. From an equivariance perspective, our proposed attention mechanism is more powerful than that of GET. For this reason, in the experiments below, the lines labeled GET correspond to retaining the GEM-CNN and EMAN convolution and using GET features as inputs.

**Relative powers.** For models using RELTAN features, we consider two choices of relative powers. Relative tangential features allow us to choose a different relative power to be used for each of the channels in the input feature (see Section 4). This increases the expressivity of the model as different relative powers induce a processing of the local geometry at a point in the mesh. We find that using multiple channels with different relative powers translates into performance improvements on the (transformed) test set.

### 7.1  Segmentation

The FAUST dataset consists of 100 (80 training, 20 test) 3-dimensional human meshes with 6890 vertices each. Nodes in each mesh are numbered so that nodes corresponding to the same "location" on the human meshes are labelled with the same number. The goal of the model is to predict, given an embedded mesh, the label for each of the nodes in the mesh.

Table 3 illustrates the performance of various gauge equivariant models on FAUST. We highlight that, rather than an overwhelmingly better prediction performance, the core advantage of using GEM-CNN or EMAN with RELTAN featuresfeatures is that the models become fully equivariant to a wide range of symmetries.

Note that models employing XYZ or GET features as input fail to be equivariant to rotations, translations and scaling transforms (Rot-Tr-Scale). This challenge is reliably overcome by RELTAN features. Using several relative powers along with with attention provides a slight boost in performance. Finally, note that solely evaluating the performance of the models on the un-transformed test set would not have been sufficient to detect the lack of equivariance to Rot-Tr-Scale transforms of the models that use XYZ or GET features.

Table 3: Means (and standard deviations over 5 seeds) of the segmentation accuracy on the FAUST dataset. No data augmentation is applied during training. Last 4 columns represent the performance of the model on the test set under *different transformations*. All models use angular biases.

| Model | Initial Features | *Accuracy* (%) | | | | |
|---|---|---|---|---|---|---|
| | | **Train** | **Test** | **Gauge** | **Rot-Tr-Scale** | **Perm** |
| **GEM-CNN** | XYZ | 99.42 (0.15) | 97.92 (0.30) | 96.90 (0.25) | **2.14** (1.49) | 97.92 (0.30) |
| | GET | 99.42 (0.15) | 98.03 (0.17) | 97.15 (0.39) | **1.47** (1.60) | 98.03 (0.17) |
| | RELTAN[0.7] | 99.69 (0.05) | 98.62 (0.06) | 98.04 (0.12) | 98.62 (0.06) | 98.62 (0.06) |
| | RELTAN[0.5, 0.7] | **99.70** (0.09) | 98.64 (0.22) | 97.99 (0.18) | 98.64 (0.22) | 98.64 (0.22) |
| **EMAN** | XYZ | 99.62 (0.09) | 98.46 (0.15) | 97.26 (0.34) | **0.02** (0.00) | 98.46 (0.15) |
| | GET | 99.60 (0.08) | 98.43 (0.17) | 97.32 (0.46) | **0.02** (0.00) | 98.43 (0.17) |
| | RELTAN[0.7] | 99.27 (1.01) | 98.13 (1.19) | 97.44 (1.26) | 98.13 (1.19) | 98.13 (1.19) |
| | RELTAN[0.5, 0.7] | 99.68 (0.00) | **98.66** (0.07) | **98.41** (0.25) | **98.66** (0.07) | **98.66** (0.07) |

## 7.2 Classification

TOSCA consists of meshes belonging to nine different classes such as cats, men, women, centaurs, etc. While figures in each class are similarly meshed, each class has a varying number of nodes and edges. The dataset consists of 80 meshes, which we uniformly split into a train set of 63 meshes and a test set of 17 meshes. The goal of the model is to predict, given an embedded mesh, the class to which the mesh belongs.

Table 4: Means (and standard deviations over 5 seeds) of the segmentation accuracy on the TOSCA dataset. No data augmentation is applied during training. Last 3 columns represent the performance of the model on the test set under *different transformations*. All models use angular biases.

| Model | Initial Features | *Accuracy* (%) | | | |
|---|---|---|---|---|---|
| | | **Train** | **Test** | **Gauge** | **Rot-Tr-Scale** |
| **GEM-CNN** | XYZ | **97.78** (2.41) | 82.35 (5.88) | 82.35 (5.88) | **12.94** (2.63) |
| | GET | 90.79 (2.84) | 82.35 (9.30) | 82.35 (9.30) | **17.65** (7.20) |
| | RELTAN[0.7] | 93.97 (4.26) | 91.76 (6.71) | 91.76 (6.71) | 91.76 (6.71) |
| | RELTAN[0.5, 0.7] | 90.16 (8.43) | 89.41 (14.65) | 89.41 (14.65) | 89.41 (14.65) |
| **EMAN** | XYZ | 47.30 (4.55) | 42.35 (20.55) | 44.71 (18.88) | **12.94** (2.63) |
| | GET | 44.13 (7.39) | 42.35 (11.31) | 41.18 (9.30) | **10.59** (2.63) |
| | RELTAN[0.7] | 92.70 (4.14) | 94.12 (4.16) | 94.12 (4.16) | 94.12 (4.16) |
| | RELTAN[0.5, 0.7] | 97.46 (4.14) | **98.82** (2.63) | **98.82** (2.63) | **98.82** (2.63) |

We do not apply permutations to the nodes in the test meshes since the different number of nodes across classes makes the implementation of this transformation cumbersome. In addition to our theoretical guarantees, and the empirical verification of permutation equivariance in the previous experiments, we do not expect node permutations to significantly affect the behavior of the models for a shape *classification* task since we use a mean pooling layer for aggregating the information across the nodes for this dataset.

We find that when using XYZ features, GEM-CNNs outperform EMANs. This seems to point to a higher sensitivity of EMANs to un-normalized data. This sensitivity is not present when using RELTAN features. We do not normalize the XYZ data in order to emphasize the fact that finding a good normalization strategy becomes unnecessary when using RELTAN features, given their scaling-invariance properties. In fact, using RELTAN features (with relative powers [0.5, 0.7]), we find EMAN to achieve the best test performance, which is also robust to all the considered transformations.

## 8  Conclusion

In this work, we propose Equivariant Mesh Attention Networks (EMAN), an attention-based model that is equi/invariant to node permutations, local gauge transformations, as well as global transformations such as rotations, translations, scalings. Our model consists of two major components: relative tangential features (RELTAN) as input types and a message passing algorithm based on a gauge equivariant attention mechanism. We also emphasize the importance of rigorous testing of the overall assembled model, since small design choices – such as biases – can result in an non-equivariant model, damaging its robustness transformations in the data. We verify the equi/invariance of our overall model theoretically and empirically. EMANs achieve competitive performance on the FAUST and TOSCA datasets, while maintaining equivariance to all the aforementioned transformations.

### Acknowledgments

Experiments on the FAUST and TOSCA datasets were performed using the HAL (Kindratenko et al., 2020) and Mila compute clusters.

This research is the result of a collaboration initiated at the London Geometry and Machine Learning Summer School 2021 (LOGML). This work utilizes resources supported by the National Science Foundation's Major Research Instrumentation program, grant #1725729, as well as the University of Illinois at Urbana-Champaign. SB was supported in part by the Department of Energy (DOE) award (DE-SC0012704). JGP is supported by the Canada CIFAR AI Chair Program and by an IVADO PhD Excellence Scholarship. FV is supported by the Engineering and Physical Sciences Research Council [EP/S021590/1] (the EPSRC Centre for Doctoral Training LSGNT — UCL and Imperial College London).

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

## Appendix

## A    Geometric Kernel Constraints in GEM-CNNs

Table 5: Solutions to the angular kernel constraint for kernels that map from $\rho_n$ to $\rho_m$, where $c_\pm = \cos((m\pm n)\theta)$ and $s_\pm = \sin((m \pm n)\theta)$. This table is directly taken from de Haan et al. (2021).

| $\rho_{\text{in}} \to \rho_{\text{out}}$ | linearly independent solutions for $K_{\text{neigh}}(\theta)$ |
|---|---|
| $\rho_0 \to \rho_0$ | $(1)$ |
| $\rho_n \to \rho_0$ | $(\cos n\theta \;\; \sin n\theta), \;\; (\sin n\theta \;\; -\cos n\theta)$ |
| $\rho_0 \to \rho_m$ | $\begin{pmatrix} \cos m\theta \\ \sin m\theta \end{pmatrix}, \begin{pmatrix} \sin m\theta \\ -\cos m\theta \end{pmatrix}$ |
| $\rho_n \to \rho_m$ | $\begin{pmatrix} c_- & -s_- \\ s_- & c_- \end{pmatrix}, \begin{pmatrix} s_- & c_- \\ -c_- & s_- \end{pmatrix}, \begin{pmatrix} c_+ & s_+ \\ s_+ & -c_+ \end{pmatrix}, \begin{pmatrix} -s_+ & c_+ \\ c_+ & s_+ \end{pmatrix}$ |
| $\rho_{\text{in}} \to \rho_{\text{out}}$ | linearly independent solutions for $K_{\text{self}}(\theta)$ |
| $\rho_0 \to \rho_0$ | $(1)$ |
| $\rho_n \to \rho_n$ | $\begin{pmatrix} 1 & 0 \\ 0 & 1 \end{pmatrix}, \begin{pmatrix} 0 & 1 \\ -1 & 0 \end{pmatrix}$ |

Message passing in GEM-CNNs de Haan et al. (2021) is defined by the equation

$$f'_p = K_{\text{self}} f_p + \sum_{q \in \mathcal{N}_p} K_{\text{neigh}}(\theta_{pq})\rho_{\text{in}}(g_{q \to p})f_q.$$

Gauge equivariance translate on the kernels $K_{\text{self}}$ and $K_{\text{neigh}}(\theta)$ as

$$K_{\text{self}} = \rho_{\text{out}}(-g)\; K_{\text{self}}\; \rho_{\text{in}}(g), \quad K_{\text{neigh}}(\theta - g) = \rho_{\text{out}}(-g)\; K_{\text{neigh}}(\theta)\; \rho_{\text{in}}(g).$$

The representation $\rho_{\text{in}}$ decomposes in irreducible components as $\oplus\rho_{n_j}$, and the representation $\rho_{\text{out}}$ as $\oplus\rho_{m_i}$. Then, the kernels $K_{\text{self}}$ and $K_{\text{neigh}}(\theta)$ are block matrices obtained by combining possible block kernels $(K_{\text{self}})_{ij}$ and $(K_{\text{neigh}}(\theta))_{ij}$ from features of type $\rho_{n_j}$ to features of type $\rho_{m_i}$. Linearly independent solutions of the kernel constraint are listed in Table 5 (derived in Weiler & Cesa (2019)). A generic kernel from features of type $\rho_n$ to feature of type $\rho_m$ is a linear combination of learnable parameters of the given basis. Table 5 also provides solutions for the kernel equations in Section 6.

## B    More on Relative Tangential Features

### B.1    The exponent $3/2$

In this section, we discuss the renormalization factor $1/N_p^{3/2}$ present in the expression of RELTAN features. We call unnormalized RELTAN features the same expression, without the renormalization factor. Unnormalized RELTAN features do not scale properly as the number of neighbors grows. In particular, we show that unnormalized RELTAN features at a node explode as the node degree increases, under reasonable assumptions. Furthermore, we make the asymphotic behavior of the size of RELTAN features explicit, as a function of the degree of the node. Finally, the rescaling provides an expression for normalized RELTAN features that does not explode nor vanish as the node degree increases.

We recall that the expression of RELTAN features is:

$$v_p(r) = \frac{1}{N_p^{3/2}} \sum_{q \in \mathcal{N}_p} \pi_p\left(\frac{q - p}{||q - p||}\right) \cdot \left[\frac{||q - p||^{r-1}}{\sum_{q' \in \mathcal{N}_p} ||q' - p||^{r-1}}\right]^{-1}$$

Note that we may rewrite $v_p$ as

$$v_p = \frac{1}{N_p^{3/2}} \cdot \pi_p \left[ \left( \sum_{q \in \mathcal{N}_p} \frac{q-p}{||q-p||^r} \right) \cdot \left( \sum_{q \in \mathcal{N}_p} ||q-p||^{r-1} \right) \right],$$

and we focus on the expression inside the square brackets.

Since we analyze the asymptotic behavior of the expected size of the relative tangent feature $v_p$, we view the vectors $q-p$, as $q \in \mathcal{N}_p$, as random vectors $X_i$, for $i = 1, \ldots, N$, where $N = N_p$ is the degree of $p$. The expression for $v_p$ becomes therefore

$$\sum_{i=1}^{N} \frac{X_i}{||X_i||^r} \cdot \sum_{j=1}^{N} ||X_j||^{r-1}.$$

We also assume that:

- The random vectors $X_i$ are independent and identically distributed (i.i.d.),

- The probability density function of the random vectors $X_i$ factors into radial and angular components,

- The angular component is a uniform distribution.

Under these assumption, the expected value of the squared norm of the vector $v_p$ is

$$\mathbb{E}\left[ \left\langle \sum_{i=1}^{N} \frac{X_i}{||X_i||^r} \cdot \sum_{j=1}^{N} ||X_j||^{r-1}, \sum_{i'=1}^{N} \frac{X_{i'}}{||X_{i'}||^r} \cdot \sum_{j'=1}^{N} ||X_{j'}||^{r-1} \right\rangle \right] = \sum_{i,j,i',j'=1}^{N} \mathbb{E}\left[ \frac{||X_j||^{r-1}||X_{j'}||^{r-1}}{||X_i||^r ||X_{i'}||^r} \left\langle X_i, X_{i'} \right\rangle \right].$$

Note that the terms with $i \neq i'$ in the sum cancel out, as the vectors $X_i$ are i.i.d., and the angular component of the distribution is uniform. The considered expected value then simplifies (only considering $i = i'$) as

$$\sum_{i,j,j'=1}^{N} \mathbb{E}\left[ \frac{||X_j||^{r-1}||X_{j'}||^{r-1}}{||X_i||^{2(r-1)}} \right].$$

This sum presents $N^3$ addends. Notice that these terms are potentially different. As we assume that the variables are i.i.d., at least $N^3 - 3N^2 + 2N$ terms – for which $i, j, j'$ are all distinct – are the same. Therefore, the sum scales with a factor of $N^3$. We introduce a factor $1/N^{3/2}$ into the original vector, so that the considered expected value neither explodes nor vanishes as $N$ becomes large.

## B.2 Proof of Lemma 4.1

We recall that RELTAN features to the mesh $\mathcal{M}$ are given by:

$$v_p^{\mathcal{M}} = \frac{1}{N_p^{3/2}} \sum_{q \in \mathcal{N}_p} \pi_p^{\mathcal{M}} \left( \frac{q-p}{||q-p||} \right) \cdot \left[ \frac{||q-p||^{r-1}}{\sum_{q' \in \mathcal{N}_p} ||q'-p||^{r-1}} \right]^{-1},$$

where we make explicit the given mesh $\mathcal{M}$.

**Equivariance under global rotations of $\mathbb{R}^3$.** Let $R \in SO(3)$ be a global rotation in $\mathbb{R}^3$, and denote by $R\mathcal{M}$ the mesh obtained by rotating $\mathcal{M}$ according to $R$. Notice that the set of neighbors $\ell \in \mathcal{N}_{Rp}$ of the node $Rp$ in the mesh $R\mathcal{M}$ is the set of points $Rq$, as $q \in \mathcal{N}_p$ for the mesh $\mathcal{M}$. Also, the normal vector $n_{Rp}^{R\mathcal{M}}$ to the mesh $R\mathcal{M}$ at the node $Rp$ is nothing but $Rn_p^{\mathcal{M}}$. Consequently,

$$\pi_{Rp}^{R\mathcal{M}} = I - n_{Rp}^{R\mathcal{M}} \left( n_{Rp}^{R\mathcal{M}} \right)^\top = I - Rn_p^{\mathcal{M}} \left( n_p^{\mathcal{M}} \right)^\top R^\top = R\pi_p^{\mathcal{M}} R^\top,$$

where we used that $RR^\top = I$. The relative tangential feature at node $Rp$ for the mesh $R\mathcal{M}$ is

$$
\begin{aligned}
v_{Rp}^{R\mathcal{M}} &= \frac{1}{N_{Rp}^{3/2}} \sum_{\ell \in \mathcal{N}_{Rp}} \pi_{Rp}^{R\mathcal{M}} \left( \frac{\ell - Rp}{||\ell - Rp||} \right) \cdot \left[ \frac{||\ell - Rp||^{r-1}}{\sum_{\ell' \in \mathcal{N}_{Rp}} ||\ell' - Rp||^{r-1}} \right]^{-1} \\
&= \frac{1}{N_p^{3/2}} \sum_{q \in \mathcal{N}_p} R\pi_p^{R\mathcal{M}} R^\top \left( \frac{R(q - p)}{||R(q - p)||} \right) \cdot \left[ \frac{||R(q - p)||^{r-1}}{\sum_{q' \in \mathcal{N}_p} ||R(q' - p)||^{r-1}} \right]^{-1} \\
&= \frac{1}{N_p^{3/2}} \sum_{q \in \mathcal{N}_p} R\pi_p^{R\mathcal{M}} \left( \frac{q - p}{||(q - p)||} \right) \cdot \left[ \frac{||q - p||^{r-1}}{\sum_{q' \in \mathcal{N}_p} ||q' - p||^{r-1}} \right]^{-1} = Rv_p^{\mathcal{M}},
\end{aligned}
$$

that is equivariance under global rotations.

**Invariance under translations of $\mathbb{R}^3$.** The argument is similar. If $x \in \mathbb{R}^3$ determines a translation, denote by $\mathcal{M} + x$ the translated mesh. The set of neighbors $\ell \in \mathcal{N}_{p+x}$ of the node $p + x$ in the mesh $\mathcal{M} + x$ is the set of points $q + x$, as $q \in \mathcal{N}_p$ for the mesh $\mathcal{M}$. Also, the normal vector $n_{p+x}^{\mathcal{M}+x}$ to the mesh $\mathcal{M} + x$ at the node $p + x$ is the original $n_p^{\mathcal{M}}$, and therefore $\pi_{p+x}^{\mathcal{M}+x} = \pi_p^{\mathcal{M}}$. Hence,

$$
\begin{aligned}
v_{p+x}^{\mathcal{M}+x} &= \frac{1}{N_{p+x}^{3/2}} \sum_{\ell \in \mathcal{N}_{p+x}} \pi_{p+x}^{\mathcal{M}+x} \left( \frac{\ell - (p + x)}{||\ell - (p + x)||} \right) \cdot \left[ \frac{||\ell - (p + x)||^{r-1}}{\sum_{\ell' \in \mathcal{N}_{p+x}} ||\ell' - (p + x)||^{r-1}} \right]^{-1} \\
&= \frac{1}{N_p^{3/2}} \sum_{q \in \mathcal{N}_p} \pi_p^{\mathcal{M}} \left( \frac{q - p}{||q - p||} \right) \cdot \left[ \frac{||q - p||^{r-1}}{\sum_{q' \in \mathcal{N}_p} ||q' - p||^{r-1}} \right]^{-1} = v_p^{\mathcal{M}},
\end{aligned}
$$

that is invariance under translations.

**Invariance under scaling of $\mathbb{R}^3$.** Again, a similar argument. Let $\lambda > 0$ be the scaling factor, determining the map $p \mapsto \lambda p$ for $p \in \mathbb{R}^3$. The set of neighbors $\ell \in \mathcal{N}_{\lambda p}$ of the node $\lambda p$ in the mesh $\lambda \mathcal{M}$ is the set of points $\lambda q$, as $q \in \mathcal{N}_p$ for the mesh $\mathcal{M}$. As above, we deduce that $\pi_{\lambda p}^{\lambda \mathcal{M}} = \pi_p^{\mathcal{M}}$. Thus, as $\lambda > 0$,

$$
\begin{aligned}
v_{\lambda p}^{\lambda \mathcal{M}} &= \frac{1}{N_{\lambda p}^{3/2}} \sum_{\ell \in \mathcal{N}_{\lambda p}} \pi_{\lambda p}^{\lambda \mathcal{M}} \left( \frac{\ell - (\lambda p)}{||\ell - (\lambda p)||} \right) \cdot \left[ \frac{||\ell - (\lambda p)||^{r-1}}{\sum_{\ell' \in \mathcal{N}_{\lambda p}} ||\ell' - (\lambda p)||^{r-1}} \right]^{-1} \\
&= \frac{1}{N_p^{3/2}} \sum_{q \in \mathcal{N}_p} \pi_p^{\mathcal{M}} \left( \frac{\lambda(q - p)}{||\lambda(q - p)||} \right) \cdot \left[ \frac{||\lambda(q - p)||^{r-1}}{\sum_{q' \in \mathcal{N}_p} ||\lambda(q' - p)||^{r-1}} \right]^{-1} \\
&= \frac{1}{N_p^{3/2}} \sum_{q \in \mathcal{N}_p} \pi_p^{\mathcal{M}} \left( \frac{q - p}{||(q - p)||} \right) \cdot \left[ \frac{||(q - p)||^{r-1}}{\sum_{q' \in \mathcal{N}_p} ||(q' - p)||^{r-1}} \right]^{-1} = v_p^{\mathcal{M}},
\end{aligned}
$$

that is invariance under scaling.

## C   Equivariant Mesh Attention Layer: proofs, multi-head, and self-contribution

### C.1   Proof of Lemma 6.1

The proof boils down to two core steps. First, the softmax-argument is invariant under gauge transformation. In addition, multiplying the matrix $\mathbf{V}_p$ (that transforms as a feature of type $\rho_{\text{out}}$) with the invariant softmax-vector produces a feature of type output. Here we provide a detailed proof.

Under a gauge transformation $g \in \mathrm{SO}(2)$, the coordinate vectors $f_p$ and $\rho_{\text{in}}(g_{q \to p})f_q$ at $p$ transform as

$$
f_p \mapsto \rho_{\text{in}}(-g)f_p, \quad \rho_{\text{in}}(g_{q \to p})f_q \mapsto \rho_{\text{in}}(-g)\rho_{\text{in}}(g_{q \to p})f_q.
$$

Also, the angle $\theta$ changes as $\theta \mapsto \theta - g$ under the same gauge transformation. Using these relations, together with the ones expressed in Equation 11, we see that $\mathbf{Q}_p$ and the $\mathbf{K}_{pq}$ transform as features of type $\rho_{\text{att}}$, while the $\mathbf{V}_{pq}$ transform as features of type $\rho_{\text{out}}$:

$$\mathbf{Q}_p \mapsto \rho_{\text{att}}(-g)\mathbf{Q}_p, \qquad \mathbf{K}_{pq} \mapsto \rho_{\text{att}}(-g)\mathbf{K}_{pq}, \qquad \mathbf{V}_{pq} \mapsto \rho_{\text{out}}(-g)\mathbf{V}_{pq}.$$

We see this explicitly, for instance, for the case of $\mathbf{K}_{pq}$:

$$\begin{aligned}
\mathbf{K}_{pq} &= K_{\text{key}}(\theta_{pq})\rho_{\text{in}}(g_{q \to p})f_q \\
&\mapsto K_{\text{key}}(\theta_{pq} - g)\rho_{\text{in}}(g^{-1})\rho_{\text{in}}(g_{q \to p})f_q \\
&= \rho_{\text{att}}(-g)K_{\text{key}}(\theta)\rho_{\text{in}}(g)\rho_{\text{in}}(g^{-1})\rho_{\text{in}}(g_{q \to p})f_q \\
&= \rho_{\text{att}}(-g)K_{\text{key}}(\theta)\rho_{\text{in}}(g_{q \to p})f_q \\
&= \rho_{\text{att}}(-g)\mathbf{K}_{pq},
\end{aligned}$$

where we made use of the constraint $K_{\text{key}}(\theta - g) = \rho_{\text{att}}(-g)K_{\text{key}}(\theta)\rho_{\text{in}}(g)$. Computations for the other cases are similar. Being obtained by column-concatenation from $\mathbf{K}_{pq}$ and $\mathbf{V}_{pq}$, the matrices $\mathbf{K}_p$ and $\mathbf{V}_p$ undergo the same transformations as well:

$$\mathbf{K}_p \mapsto \rho_{\text{att}}(-g)\mathbf{K}_p, \quad \mathbf{V}_p \mapsto \rho_{\text{out}}(-g)\mathbf{V}_p.$$

Finally, the convolutional outcome transforms as

$$\begin{aligned}
f'_p &\mapsto \rho_{\text{out}}(-g) \cdot N_p \cdot \mathbf{V}_p \cdot \text{softmax}\left( \frac{\mathbf{K}_p^\top \rho_{\text{att}}(-g)^\top \rho_{\text{att}}(-g)\mathbf{Q}_p}{\sqrt{C_{\text{att}}}} \right) \\
&= \rho_{\text{out}}(-g) \cdot N_p \cdot \mathbf{V}_p \cdot \text{softmax}\left( \frac{\mathbf{K}_p^\top \cdot \mathbf{Q}_p}{\sqrt{C_{\text{att}}}} \right) = \rho_{\text{out}}(-g) \cdot f'_p,
\end{aligned}$$

where we used the orthogonality of the representation $\rho_{\text{att}}$, namely $\rho_{\text{att}}(-g)^\top = \rho_{\text{att}}(-g)^{-1}$. In conclusion, $f'_p$ transforms as a feature of type $\rho_{\text{out}}$, and the proposed method is gauge equivariant.

## C.2  Proof of Lemma 6.2

Suppose that $R \in \text{SO}(3)$ is a global rotation of $\mathbb{R}^3$, mapping the mesh $\mathcal{M}$ to the mesh $R\mathcal{M}$. Given a feature $f$ of type $\rho_{\text{in}}$ on $\mathcal{M}$, we represent it at a point $p$ by its coordinates $f_p$, with respect to a gauge $\mathbf{E}_p$. Then, the rotation $R$ defines a feature $R_* f$ on $R\mathcal{M}$, with coordinates $(R_* f)_{Rp} = f_p$ with respect to the gauge $R\mathbf{E}_p$. Here comes the key remark: the quantities $\theta_{Rp,Rq}$ and $g_{Rq \to Rp}$ with respect to the gauge $R\mathbf{E}_p$ at $Rp$ for $R\mathcal{M}$ are precisely the quantities $\theta_{pq}$ and $g_{q \to p}$ with respect to the gauge $\mathbf{E}_p$ at $p$ for $\mathcal{M}$. Indeed, $g_{q \to p}$ can be computed as

$$g_{q \to p} = \text{atan2}\big((R_{q \to p}e_{q,2})^\top e_{p,1}, (R_{q \to p}e_{q,1})^\top e_{p,1}\big),$$

where $R_{q \to p}$ is the unique rotation of $\mathbb{R}^3$ that maps $n_q$ to $n_p$, with fixed axis $n_q \times n_p$,

For $\theta_{pq}$, instead, we notice that the angle can be written as

$$\theta_{pq} = \text{atan2}(e_{p,2}^\top \log_p(q), e_{p,1}^\top \log_p(q)),$$

where $\log_p$ is the norm-preserving discrete logarithmic map

$$\log_p(q) = ||q - p|| \frac{(I - n_p n_p^\top)(q - p)}{||(I - n_p n_p^\top)(q - p)||}.$$

Therefore, the outcome $(R * f)'_{Rp}$ of the convolution at $Rp$ for $R\mathcal{M}$ is equal to the outcome of the convolution $f'_p$ at $p$ for $\mathcal{M}$. In other words, the feature mapping defined by the convolutional update is global rotation equivariant.

A similar argument can be applied to global translation $T$ and scaling $S$: the coordinate vector of the feature do not change under $T_*$ or $S_*$, the gauge is left unchanged, and the quantities $\theta_{pq}$ and $g_{q \to p}$ are not modified. In conclusion, the convolutional step is also translation and scaling invariant. Notice that a key property, implicitly used when considering scaling, is the dependence of the kernel only on angles, and not on the radial component.

## C.3 Proof of Theorem 6.3

We prove the result for the designed Equivariant Mesh Attention models for segmentation and classification tasks, whose details are described in Appendix E.

Thanks to Lemma 6.1 and Lemma 6.2, each of the three blocks in the convolutional block is gauge and global rotation equivariant, and global translation and global scaling invariant. Therefore, the whole convolutional block satisfies the same properties, and it outputs a sum of scalar features. As operations in the dense block are defined on scalar features only, and not involving any quantities related to the geometry of the mesh, the same equi/in-variant properties hold also for the dense block. Finally, thanks to Lemma 4.1, the process of computing RELTAN features is consistent with the above properties, and the whole model is gauge invariant, global rotation equivariant, and global translation and global scaling invariant.

Regarding equivariance under permutation, it is enough to notice that all the operations involved in the convolutional block, and the process of computing RELTAN features, are permutation equivariant. Moreover, the operations in the dense block are defined node-wise, and the same operation on features is applied at each node. In conclusion, the whole model is equivariant under permutation (in the segmentation task, and invariant in the classification task).

## C.4 Multi-head

It is feasible to incorporate multi-head attention in the Equivariant Mesh Attention layer, and we present here how. However, we did not notice an improvement in performance when integrating this factor in our implementation for the considered tasks.

Choose $h$ and $d = d_{\text{model}}$ such that $C_{\text{out}} = dh$. For each $i = 1, \ldots, h$, consider projection matrices of size $d \times C_{\text{att}}$, denoted by $W^i_{\text{query}}, W^i_{\text{key}}$, and a projection matrix of size $d \times C_{\text{out}}$, denoted be $W^i_{\text{value}}$. Also, for each $i = 1, \ldots, h$ we fix a representation $\rho_i \colon \text{SO}(2) \to \mathbb{R}^d$. For these matrices, we require the gauge equivariant conditions

$$W^i_{\text{query}} = \rho_i(-g) W^i_{\text{query}} \rho_{\text{att}}(g), \qquad W^i_{\text{key}} = \rho_i(-g) W^i_{\text{key}} \rho_{\text{att}}(g), \qquad W^i_{\text{value}} = \rho_i(-g) W^i_{\text{value}} \rho_{\text{out}}(g).$$

Finally, we consider a $C_{\text{out}} \times C_{\text{out}}$ matrix $W^O$. We define the representation $\rho_{\text{diag}} \colon \text{SO}(2) \to \mathbb{R}^{C_{\text{out}}}$ by block-diagonal concatenation of the $h$ representations $\rho_i$. The gauge equivariant condition satisfied by $W^O$ is therefore

$$W^O = \rho_{\text{out}}(-g) W^O \rho_{\text{diag}}(g).$$

Then, the multihead attention outcome is defined by

$$\text{MultiHead}(\mathbf{Q}_p, \mathbf{K}_p, \mathbf{V}_p) = W^O \cdot \text{Concat}(\text{head}_1, \ldots, \text{head}_h),$$

where

$$\text{head}_i = \text{Att}(W^i_{\text{query}} \mathbf{Q}_p, W^i_{\text{key}} \mathbf{K}_p, W^i_{\text{value}} \mathbf{V}_p).$$

## C.5 Equivariant Mesh Attention Layer with Self-Contribution

Here we provide the details of the convolutional update variant including self-contribution:

$$f'_p = \alpha_{pp} K_{\text{self}} f_p + \sum_{q \in \mathcal{N}_p} \alpha_{pq} K_{\text{neigh}}(\theta_{pq}) \rho_{\text{in}}(g_{q \to p}) f_q.$$

In line with Section 6, we consider the quantities

$$\mathbf{Q}_p = K_{\text{query}} f_p, \quad \mathbf{K}_{pp} = K^{\text{self}}_{\text{key}} f_p, \quad \mathbf{K}_{pq} = K^{\text{neigh}}_{\text{key}}(\theta_{pq}) \rho_{\text{in}}(g_{q \to p}) f_q,$$
$$\mathbf{V}_{pp} = K^{\text{self}}_{\text{value}} f_p, \quad \mathbf{V}_{pq} = K^{\text{neigh}}_{\text{value}}(\theta_{pq}) \rho_{\text{in}}(g_{q \to p}) f_q.$$

Here, $K_{\text{query}}, K^{\text{self}}_{\text{key}}$, and $K^{\text{neigh}}_{\text{key}}(\theta)$ are $C_{\text{att}} \times C_{\text{in}}$ matrices, while $K^{\text{self}}_{\text{value}}$ and $K^{\text{neigh}}_{\text{value}}(\theta)$ are $C_{\text{out}} \times C_{\text{in}}$ matrices. We define $\mathbf{K}_p$ as the $C_{\text{att}} \times (N_p + 1)$ matrix obtained by concatenating as columns the column vectors $\mathbf{K}_{pp}$

and $\mathbf{K}_{pq}$, as $q$ varies. Similarly, $\mathbf{V}_p$ is the $C_{\text{out}} \times (N_p + 1)$ matrix obtained via the same procedure from the column vectors $\mathbf{V}_{pp}$ and $\mathbf{V}_{pq}$, as $q$ varies. The outcome

$$\tilde{f}_p = (N_p + 1) \cdot \mathbf{V}_p \cdot \text{softmax}\left(\frac{\mathbf{K}_p^\top \cdot \mathbf{Q}_p}{\sqrt{C_{\text{att}}}}\right)$$

is a column vector of length $C_{\text{out}}$, and in fact a feature of type output.

Gauge equivariance conditions for the matrices $K_{\text{query}}$, $K_{\text{key}}(\theta)$, and $K_{\text{value}}(\theta)$, translates as:

$$K_{\text{query}} = \rho_{\text{att}}(-g) K_{\text{query}} \rho_{\text{in}}(g),$$
$$K_{\text{key}}^{\text{self}} = \rho_{\text{att}}(-g) K_{\text{key}}^{\text{self}} \rho_{\text{in}}(g),$$
$$K_{\text{key}}^{\text{neigh}}(\theta - g) = \rho_{\text{att}}(-g) K_{\text{key}}^{\text{neigh}}(\theta) \rho_{\text{in}}(g),$$
$$K_{\text{value}}^{\text{self}} = \rho_{\text{out}}(-g) K_{\text{value}}^{\text{self}} \rho_{\text{in}}(g),$$
$$K_{\text{value}}^{\text{neigh}}(\theta - g) = \rho_{\text{out}}(-g) K_{\text{value}}^{\text{neigh}}(\theta) \rho_{\text{in}}(g).$$

## D  Node permutation equivariance of SpiralNet++

The SpiralNet++ convolution on meshes makes use of *spiral sequences* around nodes (Gong et al., 2019). Given a node, a spiral length, an orientation, and a preferred starting neighbor, the node sequence that constitutes the spiral is uniquely determined[‡]. The authors consider a fixed counter-clockwise orientation for all spiral sequences, and the choice of starting neighbor is arbitrary. With these choices, let $S(i, \lambda)$ denote the indices of the nodes belonging to the spiral sequence of length $\lambda$ starting at node $i$. The feature update follows the rule $x_i' = \texttt{MLP}\left(\big\|_{j \in S(i,\lambda)} x_j\right)$.

In Section 5, we analyze the response of SpiralNet++ to different types of transformations, including node permutation. This is equivalent to saying that the indices of the preferred neighbor choices transform according to the permutation. This may represent an issue for node permutation equivariance: since the choice of preferred neighbor is arbitrary, unless stored explicitly along with the mesh, it is not possible to guarantee that the "same" choice of starting neighbors will be made *after* having permuted the nodes. For example, consider a common mesh along with two different labelings $A$ and $B$ of its nodes, and an arbitrary choice of starting neighbors under labeling $A$. It is not possible to infer the *arbitrary* choice of starting neighbors under labeling $B$ based only on the permutation relating the change of labeling $A \to B$.

This challenge can be easily resolved by taking into account the geometry of the mesh when choosing starting neighbors: e.g., choosing the closest neighbor in the Euclidean distance as preferred neighbor.

## E  Experimental Details

**Inputs.** The input feature type XYZ is $3 \times \rho_0$; and for RELTAN features it is $\rho_0 \oplus \rho_1$, where relative tangent features are stored in the $\rho_1$ part of the feature, and the scalar $\rho_0$ is set to zero. GET features also have type $\rho_0 \oplus \rho_1$, where the $\rho_0$ component stores the projection onto the normal.

**Models.** For both segmentation and classification tasks, our models consist of two blocks: a convolution block and a dense block. The convolution block further consists of three sequential gauge equivariant residual blocks. Each residual block consists of two gauge equivariant convolutions followed by a summation of the input to the output of the block. The nature of message passing in these convolutions correspond to the choice of the model, e.g. GEM-CNN consists of convolutions of the form equation 7, whereas EMAN consists of attention mechanism equation 13.

For each model, the feature type of the input layer matches the feature type of the input features. The final layer of the sequence of residual blocks is of feature type $16 \times \rho_0$, i.e., 16 channels with only scalar features.

---

[‡]See Section 3.2 of (Gong et al., 2019) for details on the construction of the spirals.

All the intermediate feature types in the model are fixed to $16 \times (\rho_0 \oplus \rho_1 \oplus \rho_2)$. The second block consists of two dense layers. The first dense layer is of dimension $16 \times 256$ and the second dense layer maps to the target dimension followed by a softmax function. The output of the first layer is also passed through ReLU Glorot et al. (2011) and a dropout layer with parameter 0.5 Srivastava et al. (2014). The target dimension for the segmentation task on FAUST is 6890 and for classification task on TOSCA is 9. Further, in the case of classification, we also use mean pooling of the output of the first dense layer over the nodes.

**Hyperparameters.** We train using a learning rate of 0.01 for 100 epochs for FAUST segmentation tasks. In the case of TOSCA, we train for 50 epochs and use a learning rate of $2 \cdot 10^{-3}$ for GEM-CNN models, and $7 \cdot 10^{-4}$ for EMAN models. All tasks use the Adam optimizer Kingma & Ba (2015) and negative log-likelihood loss function.

## F  Equivariance vs Data Augmentation

Equivariance in machine learning has arisen as a principled alternative to data-augmentation. Section 5.1 of Thomas et al. (2018) shows that "rotation equivariance eliminates the need for rotational data augmentation" for point cloud data. Here, we show that the same applies to mesh data as well. To this end, we perform experiment on the FAUST dataset with data augmentation applied to both the training and test sets and compare the improvements brought by equivariance and data augmentation.

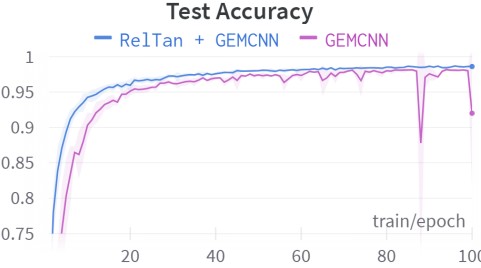

Figure 4: Effect of RELTAN features on test accuracy during training for a GEM-CNN model (averaged over 3 seeds). The results show better performance and lower variance for the model trained using RELTAN features.

Fig. 4 shows the accuracy (over 3 runs) for GEM-CNN models with initial XYZ and RELTAN features trained using roto-translation augmentations on the training set. This experiment confirms that *data augmentation* improves the generalization of non-equivariant model to unseen roto-translations: compare purple line above 90% with 2.14% test accuracy for GEM-CNN with XYZ features in Table 3.

Despite this improvement, *data augmentation is outperformed by equivariance*: the equivariant model (using RELTAN features) learns faster, has better final performance and lower variance during training.

## G  Time Comparison between EMAN and GEM-CNN

Here we provide a high-level time complexity analysis of EMAN compared to GEM-CNN. The bottleneck computations for both GEM-CNN and EMAN are expressions involving multiple matrix multiplications, of the type $K(\theta_{pq})\rho_{\text{in}}(g_{q \to p})f_q$, computed for every orientation of each edge $q \to p$. For EMAN, additionally, attention coefficients are computed for every orientation of each edge. Computation of attention coefficients involve the same type of matrix multiplication as in GEM-CNN. Therefore, we observe an increased time to process features in EMAN than in GEM-CNN. Moreover, as both EMAM and GEM-CNN time complexities are proportional to the number of such operations involved in the models, the ratio of runtimes for EMAN and GEM-CNN scales as a constant in the limit of number of edges.

In practice, from Fig. 5 we find that EMAN takes twice as much time as GEM-CNN for the FAUST dataset. However, because of the use of attention mechanism, EMAN surpasses the performance of GEM-CNN within the 30 minutes that GEM-CNN takes to complete its 100 epochs. Hence, even though EMAN has higher time-complexity, it can outperform GEM-CNN within a short window of training time.

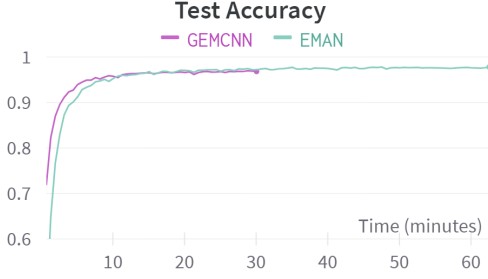

Figure 5: Wall-time comparison between GEM-CNN and EMAN. EMAN requires twice as long per epoch, due to the use of attention. However, given a fixed budget of 30 minutes, the accuracy for EMAN on the FAUST dataset surpasses that of GEM-CNN.

## H Comparison with Gauge Equivariant Transformers

In this appendix, we compare in detail our model with the Gauge Equivariant Transformer (GET) proposed by He et al. (2021). More specifically, we discuss the differences between the choices of initial features, the layer designs, and the effects of the combinations of features and model architectures.

**Comparison of features.** We recall the expression of RELTAN features, for a node $p$:

$$v_p(r) = \frac{1}{N_p^{3/2}} \sum_{q \in \mathcal{N}_p} \pi_p \left( \frac{q - p}{||q - p||} \right) \cdot \left[ \frac{||q - p||^{r-1}}{\sum_{q' \in \mathcal{N}_p} ||q' - p||^{r-1}} \right]^{-1},$$

where $N_p$ denotes the degree of node $p$. The projector $\pi_p$ onto the tangent plane $T_p\mathcal{M}$ is $I - n_p n_p^\top$, and the real number $r$ is the relative power. The vector $v_p$ is a 3D vector that belongs to $T_p\mathcal{M}$.

On the other hand, GET feature at node $p$ is nothing but the position $p$ of the node itself, considered in suitable coordinates. Assume that $\{e_{p,1}, e_{p,2}\}$ is a frame for $T_p\mathcal{M}$, and $n_p$ is the normal vector to $\mathcal{M}$ at $p$. Then, the coordinates of the GET features with respect to $\{e_{p,1}, e_{p,2}, n_p\}$, that we denote by $w_p$, are

$$w_p = \left( \langle p, e_{p,1} \rangle, \langle p, e_{p,2} \rangle, \langle p, n_p \rangle \right).$$

As noted by the authors in He et al. (2021), GET features are of type $\rho_0 \oplus \rho_1$; the $\rho_0$ component corresponds to the projection onto the normal vector, and the $\rho_1$ component to the projection onto the tangent plane. Therefore, GET features are geometric features, and formally constitute a good candidate for initial features of a gauge equivariant model, as discussed in Section 3.2.

We analyze the behavior of $v_p$ and $w_p$ when acting with a global transformation of the space $\mathbb{R}^3$. The vector $v_p$ is equivariant to rotations of the space $\mathbb{R}^3$, as stated in Lemma 4.1. This precisely means that the coefficients of $v_p$ with respect to a gauge $\{e_{p,1}, e_{p,2}\}$ are invariant under change of gauge. The same property holds for the coefficients $w_p$.

However, the situation looks different for scalings and translations. The vector $v_p$ is invariant under scalings and translations, as stated in Lemma 4.1. On the other hand, the coefficients $w_p$ are not invariant under these transformations. A scaling of a factor $\lambda$ transforms $w_p^{\mathcal{M}}$ to $w_{\lambda p}^{\lambda \mathcal{M}} = \lambda \cdot w_p^{\mathcal{M}}$, and a translation by a vector $x$ transforms $w_p^{\mathcal{M}}$ to $w_{p+x}^{\mathcal{M}+x} = w_p^{\mathcal{M}} + \left( \langle x, e_{p,1} \rangle, \langle x, e_{p,2} \rangle, \langle x, n_p \rangle \right)$ (compare with expressions in Lemma 4.1 for the behavior of RELTAN features).

Fig. 6 provides a visual interpretation of the differences between RELTAN and GET features. To keep the figure readable, we only show features at node $p$. RELTAN features remain unchanged compared to Fig. 1 regardless of the choice of global coordinates. On the other hand, a translation or scaling of the global coordinates would alter the GET features (not just their coordinate vectors, but the geometric features themselves). Our accompanying code includes a notebook with a 3D visualization of both types of features.

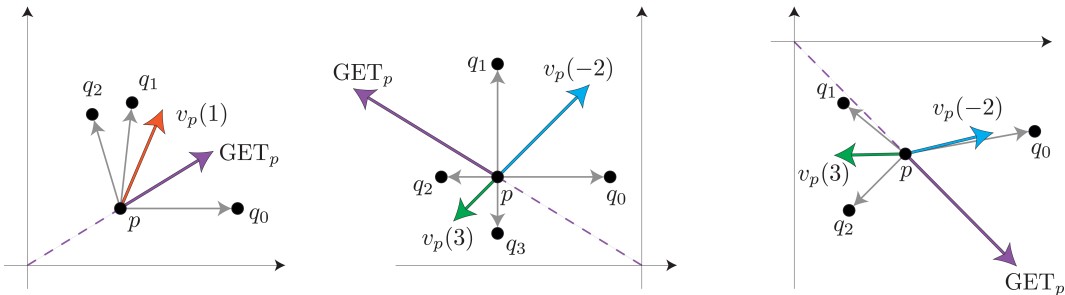

Figure 6: Visual comparison between RELTAN and GET features on planar neighborhoods. In contrast with Fig. 1, we include global coordinate axes in these plots as GET features depend on the absolute node positions (although they are in turn expressed in the local frame at node $p$).

**Comparison of the layer designs.** The GET layer is only gauge equivariant to multiples of $2\pi/N$, for a certain positive integer $N$, as He et al. (2021) consider regular representations for the hidden features. They provide a theoretical setting for an extension of regular representations to representations of the whole SO(2) (when $N$ is odd), and develop a framework to deduce the solution to the equivariant constraint. Moreover, they provide an estimation for the error in equvariance for generic rotations in SO(2). In contrast, our model is directly built on top of the convolutional kernels of GEM-CNN, for which de Haan et al. (2021) developed an architecture of precise equivariance to arbitrary rotations in SO(2).

**Comparison of features and models.** The GET model architecture applied to $w_p$ is not designed to scale properly for multiplication of a factor $\lambda$, or when adding a vector $x$ (in suitable coordinates). As a consequence, the GET model is not scaling and translation equi/in-variant.

A possible solution to this issue is a initial modification of the mesh $\mathcal{M}$, before the computation of the coefficients $w_p$. For instance, we may translate $\mathcal{M}$ so that its center of mass coincides with the origin, and scale it so that the average of the norm of the nodes $p$ is 1. This procedure annihilates the action of potential translations and scalings.

However, we argue that RELTAN features, in contrast with GET features, present another characteristic that makes them favorable for mesh processing. Their expression is *local*, as it involves the computation of relative quantities (the vectors $q - p$ and their norms) among the neighbors $q$ of a node $p$. Moreover, as discussed in Section 4, different values of the relative power $r$ detect different aspects of the neighboring disposition around $p$, and therefore of the local geometry of the mesh at $p$. Opposed to this, mere positions are prescribed in a *global* fashion, as they strictly depend on the embedding of the mesh in $\mathbb{R}^3$ and are not defined by the local geometry (that is, the neighboring nodes). Our experimental results show that the choice of RELTAN features is preferable to simple node positions.

