# OpenReview forum: "Equivariant Mesh Attention Networks"
_TMLR — Accepted by TMLR_

### Review · Reviewer_s8xe · 2022-06-18

**Summary Of Contributions:**

This paper tackles the problem of creating functions on meshes that are equivariant to transformations of the input such as rotations, scaling and permutations. The starting point of this work are guage equivariant mesh convolutional neural networks (GEM-CNN) which were among the first to design an architecture for learning functions over meshes. In that context, this paper makes two modifications:
* They propose the use of relative tangential features -- features that by design satisfy the equivariance properties and
* Propose the use of gauge equivariant attention mechanisms to parameterize the relationships that the relative tangential features have with outcomes of interest.

They show that the combination of the two results in models that are equivariant to translations, rotations, scaling, and permutations.

**Broader Impact Concerns:**

There are no major (to my knowledge) broader impact concerns with this manuscript.


**Requested Changes:**


The changes requested are in the form of clarity and writing as well as experimentation:

A] Clarity
   * I recommend clarifying the statement in the abstract that this work is the only one to look at test accuracies. As far as I can tell, Guage Equivariant CNNs also report numbers on the test set so I'm not sure what this statement is trying to convey.
   * I recommend creating a figure to visualize relative tangent features (perhaps with a side-by-side visualization against the choices made in GETs for comparison) since this is the core contribution of this work.

B] Experimentation & Related work:
   * I think the paper could have done a better work of placing the work experimentally in relation to GETs in identifying the strengths and weaknesses. One idea on this front would be to empirically evaluate the claimed rationale showing that this work outperforms GETs when meshes are translated and scaled.

**Strengths And Weaknesses:**

Strengths
* I think the idea of the relative tangent features appears to be neat, simple and in the experiments conductive effective.
* The ablation studies against the baseline show that the use of these features in both convolutional and attention based models improves robustness.

Weaknesses
* Clarity: The clarity of the idea can in several places in improved. As someone who is aware, but not an expert in neural architectures for meshes, I had to read through several other pieces of related work to understand the core contribution. I think a big part of this is that the visualizations considered in the manuscript do not go far enough in providing intuition to the reader. I think having many more visualizations of how the proposed tangential features look on very simple meshes can go a long way towards improving the readers understanding on the utility of the features. There is a statement in the abstract that this is the only work to look at accuracy on the test set which reads rather strangely.
* Experimentation and Related work: In terms of related work, the most relevant is that of He. et. al who propose guage equivariant transformers [GET]. It is difficult to attribute contribution (b) (the use of attention) solely to this work since He. et. al has already been published in the literature. In addition, the manuscript states that the key difference between this work and He et. al is that this work designs features equivariant to rotation, scale and translation (as opposed to GETs that only equivariant to rotation). I think this claim is one that can and should be tested empirically since GETs (unless I'm missing something) are the most directly related work.

---

> ### Author Response · Authors · 2022-07-05
> **Response to reviewer s8xe**
>
> We thank the reviewer for their detailed comments on the paper and for appreciating the novelty of relative tangent features and the robustness of the EMAN architecture as shown by the performed ablation studies. We also value the reviewer’s suggestions on how to address the weaknesses in our submission.
>
> ### A. Clarity
>
> *“I recommend clarifying the statement in the abstract that this work is the only one to look at test accuracies. As far as I can tell, Guage Equivariant CNNs also report numbers on the test set so I'm not sure what this statement is trying to convey.”*
>
> The statement mentioned in the abstract was:
>
> > “Moreover, previous implementations have not always applied these symmetry transformations to the test dataset. This inhibits the ability to determine whether the model attains the claimed equivariance properties. [..]. We carry out experiments on the FAUST and TOSCA datasets, and apply the mentioned symmetries to the test set only.”
>
> The goal of this sentence was to convey the importance of applying symmetry transformations during evaluation to verify that the desired equivariances are indeed achieved by the model in practice.
>
> It is indeed the case that GEM-CNNs report test performance on the *untransformed* test set.  However, previous papers did not use transformations such as gauge transformations or rotation-translation-scaling transformations on the test set to verify equivariance in practice. For instance, in the case of experiments on the FAUST dataset, the gauges chosen in the (untransformed) test set are often the same as in the train set.
>
> We agree with the reviewer that our phrasing in the abstract was confusing. The space-constrained setting of the abstract was not a suitable place to carry out this subtle discussion. We have thus decided to remove this statement from the abstract, and transfer it to Section 5 “Verifably Equivariant Message Passing”. We hope that the supporting information contained in Tables 1 and 2, along with the revised writing will improve the quality of our communication on the importance of applying transformation during evaluation.
>
> ---
>
> *“I recommend creating a figure to visualize relative tangent features (perhaps with a side-by-side visualization against the choices made in GETs for comparison) since this is the core contribution of this work.”*
>
> We have also adjusted the mathematical presentation of relative tangential features to an equivalent, but more evocative formula. This revision provides greater intuition on the behavior of relative tangential features.
>
> Thank you for suggesting these additional visualizations. We have added an extra visualization of RelTan features for the case $r=1$. We have also added a copy of this plot in Appendix H, where we also include GET features. However, it is challenging to faithfully represent the 3D features in flat paper in a way that the reader can easily visualize.
>
> Therefore, we have created an accompanying notebook with a 3D visualization of RelTan and GET feature on simple meshes. The notebook is intuitive and it is very easy to modify the input mesh. This visualization also includes a way to interactively adjust the value of the relative power $r$. We believe that this interactive visualization can provide the interested reader with much more intuition on the behavior of RelTan and GET features, than we could convey in text.
>
> ---
>
> ### B] Experimentation & Related work:
>
> *“I think the paper could have done a better work of placing the work experimentally in relation to GETs in identifying the strengths and weaknesses. One idea on this front would be to empirically evaluate the claimed rationale showing that this work outperforms GETs when meshes are translated and scaled.”*
>
> We thank the reviewer for suggesting a more thorough comparison to the work of He et al. (2021). We have addressed this in multiple ways:
> -   Our revised related work includes a more comprehensive description of GETs. We discuss how our proposed RelTan features and attention mechanism address some of their shortcomings.
> -   Since the attention mechanism proposed in GET only achieves gauge equivariance for specific angles of rotation, we decided to adopt our attention mechanism, which is equivariant to arbitrary gauge changes. We extend all our experiments by considering GEM-CNN and EMAN models with GET features as inputs.
> -   Although models with GET features achieve competitive performance on the training and test sets, they do not achieve equi/in-variance to rotation-translation-scaling transformations of the meshes.
> -   We have included a detailed comparison between our work and GET in Appendix H, including discussions on the choice of features, layer designs, and the effects of the combinations of these features and model architectures.
>
> Gauge Equivariant Transformers are indeed highly related to our work. We hope that the reviewer will find our revision satisfactory in contextualizing and comparing to this prior work.

---

### Review · Reviewer_huat · 2022-06-20

**Summary Of Contributions:**

This paper addresses the question of how to design equivariant graph convolutional networks for 3D meshes.

The proposed method first construct relative tangential features that transform absolute node positions in 3d coordinates to features based on tangential vectors that are not tied with any absolute position. To learn on these relative tangential features, the submission proposes to employ a prior method, gauge equivariant convolution. The submission extends gauge equivariant convolution with a gauge equivariant attention mechanism. Furthermore, the submission designs a better way to include convolution bias (anguilar bias) that ensures gauge equivariance.

The submission evaluates the proposed method on two benchmarks, FAUST and TOSCA, and showed that the proposed method is more robust to different transformations applied onto the test data.

**Broader Impact Concerns:**

I do not have concerns with the broader impact of this paper,

**Requested Changes:**

- Figure 1 is not referred to or explained in the main text and I am also not sure I understand the figure well. Can the authors clarify what this figure is intended to mean?
- To clarify on the experiment setup, is the parallel transport also applied to the baseline method GEM-CNN?

**Strengths And Weaknesses:**

Strengths:
- The proposed method to be novel
- The proposed method doesn't not require modifications (e.g. data augmenation) during training to achieve equivariance
- The proposed method is empirically better than previous methods in the experimental evaluation. The usage of relative tangential features is very effective.

Weak:
I do not have major concerns with the submission besides I think the presentation could be made clearer. I detail my suggestions in the next section.

---

> ### Author Response · Authors · 2022-07-05
> **Response to reviewer huat**
>
> Thank you for your careful reading of our work. We appreciate the reviewer’s positive comments.
>
> ---
>
> *“Figure 1 is not referred to or explained in the main text and I am also not sure I understand the figure well. Can the authors clarify what this figure is intended to mean?”*
>
>
> The goal of this figure is to illustrate the gauge equivariance property of our attention mechanism. The combination of a gauge equivariant kernel with the scalar features corresponding to the attention coefficients enables us to create a gauge equivariant model. This is precisely the content of Lemma 6.1. The figure illustrates how these two components “fit together” in a way that behaves properly under gauge transformations.
>
> We have adjusted the placement of this figure in the paper. It is now part of the section on Equivariant Attention, right next to Lemma 6.1. We hope that displaying the figure later will ensure that the reader is familiar with the notation and sufficiently contextualized with the importance of Lemma 6.1.
>
> ---
>
> *“To clarify on the experiment setup, is the parallel transport also applied to the baseline method GEM-CNN?”*
>
> Yes, parallel transport is a crucial component of the GEM-CNN architecture. Their use of geometric features requires parallel transport to ensure a “coherent” aggregation of features when applying the convolution operator. A detailed explanation of the use of parallel transport in GEM-CNNs can be found in Section 4 of de Haan et al. (2021).

---

### Review · Reviewer_AwSq · 2022-06-27

**Summary Of Contributions:**

This paper proposes a equivariant mesh attention (EMAN) network that is gauge equivariant and invariant with respect to rotation, translation, and scaling. The two main ingredients are the relative tangent feature and a combination of attention and GEM-CNN. The equivariance and invariance properties are verified theoretically and empirically. Experiment results show that the proposed EMAN performs well on FAUST and TOSCA datasets.

**Broader Impact Concerns:**

I do not see any such concerns.

**Requested Changes:**

Please address the weaknesses above and revise the paper is necessary.

**Strengths And Weaknesses:**

Strengths:
1. EMAN seems to be novel.
2. The computational cost is compatible to GEM-CNN.

Weaknesses:
1. The mathematical writing can be improved. For example, the notation of a tangent plane should be defined before its first appearance (e.g., in the first sentence of the second paragraph of Section 2.1), $N_p$ is not defined (though can be deduced), the meanings of * in $R_*$ and $T_*$ are unclear, the equation between (9) and (10) should be written in terms of $K_{pq}$ and $V_{pq}$, and "cat" in Algorithm 1 lacks a definition.
2. The last paragraph of Section 3 emphasizes that (7) does not involve self-contribution but does not provide any reason except that the performance is already satisfying. How is the performance if self-contribution is included?
3. The insight provided in Appendix B.1 is unclear. What is the reason of the analogy between q-p and a random vector?
4. I do not see any principled way to rule of thumb to tune the relative power.

---

> ### Author Response · Authors · 2022-07-05
> **Response to reviewer AwSq**
>
> We thank the reviewer for their detailed review and positive comments about our work. We are also thankful to the reviewer for pointing out the weaknesses of the paper and providing suggestions for improving the quality of the paper. Up next, we address all the issues raised by the reviewer.
>
> ---
>
> *“The mathematical writing can be improved” - Clarify notation.*
>
> Thank you for your careful examination of our manuscript. We agree with the reviewer that there were some missing definitions on notation, as well as some ambiguous notation.
> In the revised version of the paper, we have clarified all the notational concerns raised by the reviewer.
>
> ---
>
> *“The last paragraph of Section 3 emphasizes that (7) does not involve self-contribution but does not provide any reason except that the performance is already satisfying. How is the performance if self-contribution is included?”*
>
>
> The main reason for not adding the self contribution is that official the implementation of GEM-CNN does not use this component in their architecture. In order to keep a fair comparison between both methods, we avoided using it. Moreover, since the current architecture already achieves excellent classification accuracy, we did not consider adding the self contribution to our implementation.
> We have added these considerations to the explanation for not using self-contributions in the revised version of the paper.
>
> ---
>
> *“The insight provided in Appendix B.1 is unclear. What is the reason of the analogy between q-p and a random vector?”*
>
> The goal of this section is to show that under mild conditions, unnormalized RelTan features scale poorly with a growing number of neighbors. The choice of the factor $N_p^{3/2}$ is a result of a theoretical exercise to ensure “appropriate” behavior of the RelTan features under the mentioned conditions.
>
> Our interpretation of $q-p$ as a random vector is purely done with the aim of providing a simple setting in which the unnormalized RelTan features would diverge as the number of neighbors increases. We make these assumptions more explicit in the new version.
>
> We have edited the content of Appendix B.1 to contextualize the reader and provide a clearer motivation for this section. For convenience, we provide an extract of the new text here:
>
> > In this section, we discuss the renormalization factor $N_p^{3/2}$ present in the expression of RelTan features. We call unnormalized RelTan features the same expression, without the renormalization factor. Unnormalized RelTan features do not scale properly as the number of neighbors grows. In particular, we show that unnormalized RelTan features at a node explode as the node degree increases, under reasonable assumptions. Furthermore, we make the asymptotic behavior of the size of RelTan features explicit, as a function of the degree of the node. Finally, the rescaling provides an expression for normalized RelTan features that do not explode nor vanish as the node degree increases.
>
> ---
>
> *“I do not see any principled way to rule of thumb to tune the relative power.”*
>
> Thank you for raising this concern. The relative power is indeed a hyperparameter introduced by our approach. Although we lack a precise “rule of thumb” for how to select this hyperparameter, we have included a series of comments regarding the choice of $r$ in practice.
>
> > **Selecting relative powers.** [...] Different values of the relative power $r$ provide different perspectives on the local geometry of the mesh $\mathcal{M}$ around the node $p$. Balancing the importance of the directional and distance components may depend on domain-specific properties of the data. Moreover, multiple relative powers can be simultaneously used for capturing information about the local neighborhoods “at different scales”. Which of these scales is most relevant for the task at hand can be in turn learned as part of the optimization of the model weights during training. Note, however, that this strategy increases the number of parameters in the model. Hence, one should use enough relative powers that can capture rich information about the nodes while not being computationally wasteful. In our experiments, choosing two relative powers simultaneously provided desirable performance.
>
> We have also adjusted the mathematical presentation of relative tangent features to an equivalent, but more evocative formula; and included a greater discussion on the influence of the relative power. We hope these comments are can guide the choice of an appropriate (group of) relative power(s) in new applications.

---

> > ### Comment · Reviewer_AwSq · 2022-07-25
> > **Leaning accept**
> >
> > The revision is much more readable. The authors should try their best to polish their drafts before submitting them to journals afterwards.
> >
> > Two typos in the revision I noticed:
> > - p. 3: It seems that $\{ F \in p \}$ after (1) should be $\{ F \ni p \}$.
> > - p. 3: take the mesh structure *into* consideration…

---

### Decision · Action_Editors · 2022-08-08

**Recommendation:** Accept with minor revision

**Comment:**

This paper proposes a equivariant mesh attention (EMAN) network that is gauge equivariant and invariant with respect to rotation, translation, and scaling, based on relative tangent feature and a combination of attention and GEM-CNN. Three reviewers provided in-depth reviews of the submission, and they all see the potential value of the work. On the other hand, they also raised a number of minor concerns, including:

1) The mathematical writing can be improved.
2) The principled way of tuning the relative power is lacking
3) The placement of some figures and the necessary explanation of them can be optimized.
4) The description of some experimental details are not cleary enough, and some experimental settings (including the comparison with GET) can be improved.

Overall speaking, the authors have done a good job in addressing the concerns from the reviewers. Some reviewers have explicitly confirmed that the revisions have addressed their concerns. Although some other reviewers have not made their confirmations, as far as I can see, most of the issues have been removed too. In this way, I would recommend ACCEPTING the manuscript subject to some minor revisions. When submitting the final version of the paper, I would like to ask the authors to proof-read the paper carefully to address any remaining writing issue.

---

> ### Author Response · Authors · 2022-08-27
> **Response to Decision by Action Editor**
>
> We thank the Action Editor for their decision on our submission. We have uploaded the camera-ready version with the requested corrections.